# Multi-step recognition of potential 5' splice sites by the *Saccharomyces cerevisiae* U1 snRNP

**Sarah R Hansen[1,2], David S White[1], Mark Scalf[3], Ivan R Corrêa[4], Lloyd M Smith[3], Aaron A Hoskins[1,2,3]***

[1]Department of Biochemistry, University of Wisconsin–Madison, Madison, United States; [2]Integrated Program in Biochemistry, University of Wisconsin–Madison, Madison, United States; [3]Department of Chemistry, University of Wisconsin–Madison, Madison, United States; [4]New England Biolabs, Ipswich, United States

**Abstract** In eukaryotes, splice sites define the introns of pre-mRNAs and must be recognized and excised with nucleotide precision by the spliceosome to make the correct mRNA product. In one of the earliest steps of spliceosome assembly, the U1 small nuclear ribonucleoprotein (snRNP) recognizes the 5' splice site (5' SS) through a combination of base pairing, protein-RNA contacts, and interactions with other splicing factors. Previous studies investigating the mechanisms of 5' SS recognition have largely been done in vivo or in cellular extracts where the U1/5' SS interaction is difficult to deconvolute from the effects of *trans*-acting factors or RNA structure. In this work we used colocalization single-molecule spectroscopy (CoSMoS) to elucidate the pathway of 5' SS selection by purified yeast U1 snRNP. We determined that U1 reversibly selects 5' SS in a sequence-dependent, two-step mechanism. A kinetic selection scheme enforces pairing at particular positions rather than overall duplex stability to achieve long-lived U1 binding. Our results provide a kinetic basis for how U1 may rapidly surveil nascent transcripts for 5' SS and preferentially accumulate at these sequences rather than on close cognates.

**\*For correspondence:**
ahoskins@wisc.edu

## Editor's evaluation

This study extends previous work from the same group on the mechanism of 5' splice site recognition by the U1 snRNP using co-localization single-molecule spectroscopy. Compelling experimental and analytical approaches yielded three important conclusions: (1) the association of the U1 snRNP with the 5' splice site is largely determined by the snRNP itself and does not require other splicing factors; (2) sequence features of the 5' splice site determine whether a short-lived complex with U1 dissociates or transitions into a longer-lived, "productive" complex, potentially mediated by stabilized contacts with U1 associated proteins; and (3) the ability to form the longer-lived complex cannot be accurately predicted by base-pairing potential alone, as presumed by many predictive algorithms. This work will be of interest to colleagues in the splicing field as well as to others in fields where nucleic acid recognition by snRNPs plays a major role.

## Introduction

In eukaryotes, the introns of precursor messenger RNA (pre-mRNA) must be identified and removed with nucleotide precision by the spliceosome to produce mRNA (*Wahl et al., 2009*). The junction between an intron and the upstream exon is marked by the 5' splice site (5' SS) sequence, a motif that is essential for assembly of the spliceosome and both catalytic steps of splicing. Though the 5' SS is

marked by a conserved consensus sequence (5′-GUAUGU in yeast, 5′-GURAG in humans), only the first two nucleotides are nearly invariant (99% GU, <1% GC) as they are necessary for catalysis (*Fouser and Friesen, 1986*; *Konarska, 1998*; *Parker and Siliciano, 1993*; *Roca et al., 2013*; *Vijayraghavan et al., 1989*; *Wilkinson et al., 2017*). The other positions are degenerate, especially in the human genome where more than 9000 variants of the −3 to +6 region of the 5′ SS are utilized (*Carmel et al., 2004*; *Roca et al., 2013*). Despite this degeneracy, the precise determination of exon-intron boundaries is essential to healthy cellular function. An estimated 50% of all disease-related point mutations alter splicing in some way, with 14% of all disease-related point mutations occurring at splice sites (*Soemedi et al., 2017*).

U1 small nuclear ribonucleoprotein complex (snRNP) is responsible for 5′ SS selection during the earliest steps of spliceosome assembly (*Lacadie and Rosbash, 2005*; *Rosbash and Séraphin, 1991*; *Ruby and Abelson, 1988*). The 5′ SS consensus sequence is complementary to the 5′ end of U1 small nuclear RNA (snRNA) (*Lerner et al., 1980*). Since the first 10 nucleotides of U1 snRNA (splice site recognition sequence [SSRS]) are perfectly conserved between yeast and humans, this implies a conserved mechanism of 5′ SS selection determined, in part, by base pairing (*Rosbash and Séraphin, 1991*). However, the degeneracy of certain positions within the 5′ SS consensus shows that SSRS/5′ SS duplexes often form with less than complete complementarity, and a subset of SSRS/5′ SS interactions can even occur with noncanonical registers (*Roca and Krainer, 2009*). Additionally, there are many sequences which have a high degree of complementarity to U1 but are not utilized as splice sites (pseudo 5′ SS) or only used when nearby canonical 5′ SS are inactivated (cryptic 5′ SS) (*Roca et al., 2013*). Together these observations show that base-pairing strength with the U1 SSRS alone cannot predict 5′ SS usage. Since the 5′ SS must be transferred from the U1 to the U6 snRNA for splicing to occur, spliced mRNA formation is a convolution of multiple 5′ SS recognition events (*Brow, 2002*). As a result, it is difficult to determine how U1 SSRS/5′ SS interactions change between different sequences based on analysis of mRNAs.

Structural biology of both *Saccharomyces cerevisiae* (yeast) and human U1 snRNP has revealed how snRNP proteins could play key roles in 5′ SS recognition in addition to base pairing with the SSRS. In crystal structures of human U1 snRNP bound to a 5′ SS-containing RNA oligonucleotide (oligo), the conserved U1-C protein (Yhc1 in yeast) contacts the SSRS/5′ SS duplex in the minor groove at the pairing site between the nearly invariant 5′ SS G(+1) and U(+2) nucleotides with the snRNA (*Kondo et al., 2015*; *Pomeranz Krummel et al., 2009*). Similarly, in cryo-EM structures containing yeast U1 snRNP, Yhc1 contacts the SSRS/5′ SS duplex, also near G(+1), while a second yeast splicing factor, Luc7, contacts the snRNA strand opposite (*Figure 1A and B*; *Bai et al., 2018*; *Li et al., 2019*; *Plaschka et al., 2018*). The proximity of Yhc1 and Luc7 to the SSRS/5′ SS duplex are also consistent with genetic data supporting roles for these proteins in 5′ SS recognition (*Chen et al., 2001*; *Fortes et al., 1999*; *Schwer and Shuman, 2015*; *Schwer and Shuman, 2014*). Filter-binding competition assays using a reconstituted human U1 snRNP showed that U1-C contributes to the affinity and specificity of U1 for 5′ SS RNA oligos (*Kondo et al., 2015*). However, these assays are difficult to interpret with respect to a mechanism of 5′ SS discrimination since it is unclear if equilibrium was reached during the experiment (*Jarmoskaite et al., 2020*), the assay was limited in its ability to directly detect interactions with non-consensus 5′ SS, and it provided no information on how or if the kinetics of U1 interactions differed between different 5′ SS RNAs. Thus, it is unknown if recognition of 5′ SS originates from U1′s failure to bind mismatched RNAs or due to a selection event occurring after association.

Colocalization single-molecule spectroscopy (CoSMoS) has previously been used to study the kinetics of both yeast and human U1/RNA interactions in cell extracts (*Braun et al., 2018*; *Hoskins et al., 2011*; *Larson and Hoskins, 2017*; *Shcherbakova et al., 2013*). In all cases, short- and long-lived, 5′ SS-dependent interactions were observed between U1 and immobilized pre-mRNAs. In previous work from our laboratory with yeast U1 in whole cell extract (WCE), we showed how the populations of short- and long-lived interactions as well as their lifetimes can vary depending on the presence of a consensus or weak (containing additional mismatches) 5′ SS or due to mutation of Yhc1 (*Larson and Hoskins, 2017*). These interactions were also strongly influenced by the presence or absence of *trans*-acting factors that bind elsewhere on the pre-mRNA, including the nuclear cap-binding complex (CBC) or the branch site bridging protein (BBP)/Mud2 complex, that together with U1 form the yeast E complex spliceosome or commitment complex (*Larson and Hoskins, 2017*; *Séraphin and Rosbash, 1991*; *Seraphin and Rosbash, 1989*). Our results were consistent with a two-step mechanism for 5′ SS

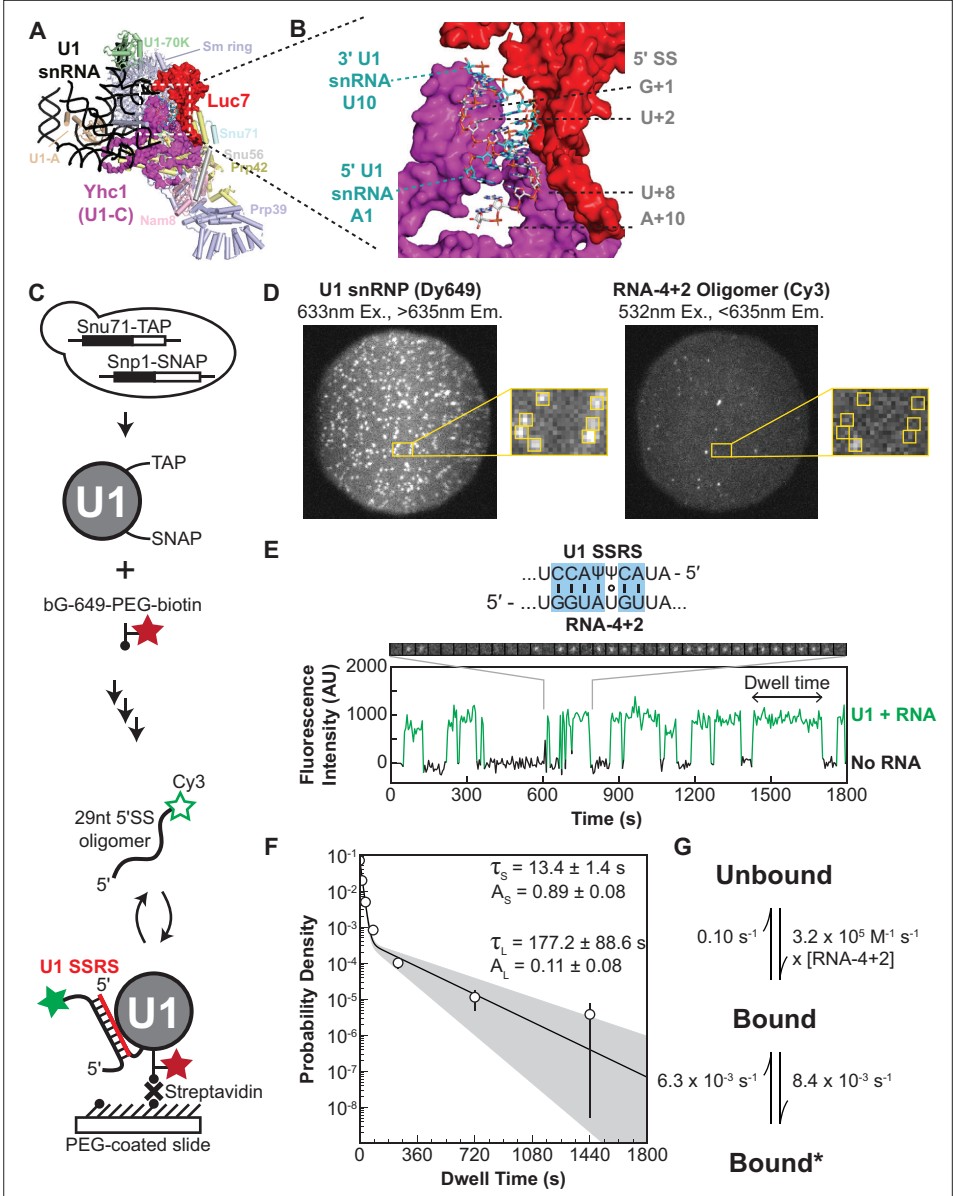

**Figure 1.** Immobilized yeast U1 small nuclear ribonucleoprotein (snRNP) forms reversible short- and long-lived interactions with a 5' splice site (5' SS) oligo. (**A**) Cryo-EM structure of the yeast U1 snRNP obtained as part of the spliceosome A complex (PDB 6G90). U1 proteins are labeled and shown as either cartoons or spacefill (Yhc1 and Luc7). The U1 snRNA backbone is shown as a black ribbon. (**B**) Expanded view of the region within the dotted box in panel (**A**) showing the cleft formed by Yhc1 (purple) and Luc7 (red) that binds the U1 SSRS/5' SS duplex. Nucleotides at the 5' and 3' ends of the of the SS and splice site recognition sequence (SSRS) are labeled. (**C**) Preparation of purified, fluorescently labeled yeast U1 snRNP using SNAP and TAP tags. In single-molecule experiments, U1 snRNP is immobilized to the slide surface and its interactions with Cy3-labeled RNA oligomers are observed using colocalization single-molecule spectroscopy (CoSMoS). The U1 SSRS that binds to the oligo is shown in red. (**D**) Images showing individual U1 snRNP molecules tethered to the slide surface (left field of view, FOV) and colocalized Cy3-labeled RNA-4+2 molecules (right FOV). Each FOV is ~50 µm in diameter. (**E**) Representative fluorescence trajectory of changes in Cy3 intensity (green) due to oligo binding to a single immobilized U1 molecule. RNA-binding events appear as spots of fluorescence in the recorded images (see inset). Also shown is the predicted pairing interactions (blue) between the RNA-4+2 oligo and the U1 SSRS. (**F**) Probability density histogram of dwell times for the RNA-4+2 oligo (*N*=367) and the fitted parameters of the data to an equation containing two exponential terms; the shaded region represents the uncertainty associated with the parameters. The dwell times are plotted as binned values, with bins values chosen that adequately

*Figure 1 continued on next page*

*Figure 1 continued*

represent the underlying distribution for visualization. The error bars of each bin are computed as the error or of a binomial distribution. The ordinate values are plotted on a log-scale to highlight the difference in short- and long-lived components (see Methods for more details). (**G**) Kinetic model with optimized rate constants describing the interaction between U1 snRNP and RNA-4+2. In this scheme, 'Bound' and 'Bound*' states correspond to the short- and long-lived bound time constants observed in the dwell time analysis, respectively.

The online version of this article includes the following source data and figure supplement(s) for figure 1:

**Source data 1.** Sequences and predicted thermodynamic stabilities of RNA oligos.

**Source data 2.** Fit parameters for data collected at fivefold increased frame rate.

**Source data 3.** Results from hidden Markov modeling of binding data for RNA-4+2.

**Figure supplement 1.** Mass spectrometry analysis of purified U1 samples.

**Figure supplement 2.** Dideoxy sequencing of the purified U1 small nuclear ribonucleoprotein (snRNA) and activity assay.

**Figure supplement 2—source data 1.** Uncropped phosphorimage of the dideoxy sequencing gel shown in *Figure 1—figure supplement 2A*.

**Figure supplement 2—source data 2.** Uncropped phosphorimage of the precursor messenger RNA (pre-mRNA) splicing assay shown in *Figure 1—figure supplement 2B*.

**Figure supplement 3.** Observed U1-binding events are sequence-dependent.

**Figure supplement 4.** U1-binding events at 1 frame per second.

recognition by U1 that involves reversible formation of an initial weakly bound complex with RNA that can transition to a more stably bound state, as was proposed previously by others (*Du et al., 2004*; *McGrail and O'Keefe, 2008*). Yet, neither our prior experiments nor those from other laboratories could exclude roles for other, non-U1 splicing factors present in the WCE in this process or potential influence of pre-mRNA structure on the observed kinetics.

In this study, we use CoSMoS to directly observe how individual yeast U1 snRNP molecules interact with short RNA oligos. The short- and long-lived interactions observed in cell extracts with large pre-mRNA substrates are also observed when purified U1 snRNP binds cognate RNAs providing direct evidence for these kinetic features being inherent to 5′ SS recognition. By using RNA oligos with varying base-pairing strength to the snRNA as well as with different locations and types of mismatches, we show that 5′ SS recognition leading to long-lived complexes occurs subsequent to binding. RNAs with limited pairing to the SSRS are released quickly after association while those with extended complementarity and pairing at certain positions are more likely to be retained and form long-lived complexes. Significantly, formation of long-lived U1/RNA complexes does not always correlate with the predicted thermodynamic stabilities of the SSRS/5′ SS RNA duplexes, which likely reflects the importance of snRNP proteins in the process. We propose that U1 uses a multi-step kinetic pathway to discriminate between RNAs and that formation of long-lived complexes is dependent on multiple factors that together favor U1 accumulation on introns competent for splicing.

## Results

### U1 forms short- and long-lived complexes with RNAs containing a 5′ SS sequence

Since we wished to study U1/5′ SS interactions in the absence of *trans*-acting factors, we first developed a protocol for purifying fluorophore-labeled U1 snRNP from yeast extract. We genetically encoded a tandem affinity purification (TAP) tag on the U1 protein Snu71 and a SNAP-tag on the U1 protein Snp1 in a protease-deficient, haploid yeast strain (*Figure 1C*). TAP-tagged Snu71 has previously been used to purify U1 snRNP (*van der Feltz and Pomeranz Krummel, 2016*; *Rigaut et al., 1999*), and SNAP-tagged Snp1 has been used to fluorescently label and visualize U1-binding events by single-molecule fluorescence in WCE (*Hoskins et al., 2011*; *Larson and Hoskins, 2017*). Extracts were prepared from the dual-tagged strain, and U1 snRNP purified using published protocols (*van der Feltz and Pomeranz Krummel, 2016*). Fluorophore labeling was carried out concertedly with TEV protease cleavage of the TAP tag, and excess fluorophore was removed during calmodulin affinity

purification. In these experiments, a tri-functional SNAP-tag ligand containing a Dy649 fluorophore, biotin, and benzyl-guanine leaving group (*Smith et al., 2013*) was used to simultaneously fluorophore label and biotinylate U1 on the Snp1 protein.

Purified U1 was characterized by mass spectrometry, and samples contained all known U1 components (*Figure 1—figure supplement 1*). Only a small number of peptides from other yeast splicing factors were identified, and these were not present in all replicates. Dideoxy sequencing of the isolated U1 confirmed the presence of the snRNA and SSRS, and the purified U1 was able to restore the splicing activity of WCE in which the endogenous U1 snRNA was degraded by targeted RNase H cleavage of the snRNA (*Du and Rosbash, 2001*; *Figure 1—figure supplement 2*). Together the data support purification of functional U1 particles.

For substrates, we designed a set of Cy3-labeled, 29 nucleotide (nt)-long RNA oligonucleotides with varying degrees of complementarity to U1 (*Figure 1—source data 1*). The RNAs are based on the RP51A pre-mRNA 5′ SS sequence, a well-studied splicing substrate (*Hoskins et al., 2011*; *Larson and Hoskins, 2017*; *Rymond and Rosbash, 1985*) and are identical except for substitutions within the 5′ SS region. Importantly, the RNAs contain the entire region known to cross-link with the U1 snRNA (*McGrail and O'Keefe, 2008*) and all U1-interacting nt that could be modeled into cryo-EM densities of the spliceosome E and A complexes (the ACT1 intron stem loop observed in E complex being an exception) (*Li et al., 2019*; *Plaschka et al., 2018*). The RNAs also contain all the sites shown to cross-link with U1 snRNP proteins except for non-conserved poly-U tracts located downstream of the 5′ SS (+27–46) that likely interact with the RRM domains of Nam8 (*Plaschka et al., 2018*; *Puig et al., 1999*; *Zhang and Rosbash, 1999*). We omitted this region to avoid potential interferences from 5′ SS-independent RRM/RNA interactions and folding of larger RNA substrates into structures that could compete with U1 interactions. The RNAs are predicted to have minimal stable secondary structure by mFold (*Zuker, 2003*) and range from limited complementarity with U1 (no more than two predicted contiguous base pairs; *Figure 1—source data 1*, RNA-control or RNA-C) to a maximum of 10 contiguous potential base pairs (RNA-10).

We immobilized the purified U1 snRNP with streptavidin on a passivated and biotinylated glass slide (*Salomon et al., 2015*) and readily observed single spots of fluorescence from the Dy649 fluorophore upon excitation at 633 nm (*Figure 1D*). When a 29-nt RNA oligo containing a consensus 5′ SS and Cy3 fluorophore (RNA-4+2) was introduced, spots of Cy3 fluorescence began to transiently appear on the surface (*Figure 1D and E*). The spots of Cy3 fluorescence colocalized with the immobilized U1 molecules, and spots repeatedly appeared and disappeared from the same U1 molecule. This is consistent with multiple rounds of binding and release of the RNA-4+2 oligo during the experiment. As a control, we added a Cy3-labeled oligo which lacked any significant complementarity to U1 (*Figure 1—source data 1*, RNA-C). We observed few Cy3 signals on the surface, and the event density (frequency of colocalized binding events) of RNA-C was 40-fold less than that of RNA-4+2 (*Figure 1—figure supplement 3*). While it is possible that non-specific interactions between RNA-C and U1 occurred too rapidly for us to detect, the large differences in event density between RNA-C and RNA-4+2 indicate that the vast majority of the detected binding events represent sequence-specific interactions.

RNA-4+2 dwell time distributions were analyzed using maximum likelihood methods (*Kaur et al., 2019*). We found that two exponential components were required to explain our observations (*Figure 1F*). This is consistent with the appearance of both short- and long-lived binding events observed in the time trajectories of single U1 molecules (*Figure 1C*). The short-lived kinetic parameter ($\tau_s$) was ~13 s with an amplitude of 0.89 (where amplitude reflects the proportion of this time constant across the entire distribution), while the long-lived kinetic parameter ($\tau_L$) was much larger (~177 s) but with a smaller amplitude (0.11). When we increased the frame rate fivefold, we observed a similar distribution of signals that yielded similar fit parameters (*Figure 1—figure supplement 4* and *Figure 1—source data 2*). This indicates that $\tau_s$ values of about 10 s can be effectively measured using our imaging protocol but does not preclude the presence of binding events with sub-second lifetimes.

To further explore the functional dynamics underlying our observations, we compared the likelihood of several kinetic models containing either one or two bound states using hidden Markov modeling (*Figure 1G*, *Figure 1—source data 3*; *Qin et al., 2000*; *Nicolai and Sachs, 2013*). A model featuring an initial short-lived RNA association followed by a transition to a long-live state was the

most likely to explain our data, consistent with our dwell time analysis suggesting two bound populations. The most direct interpretation of this finding is that the U1 snRNP undergoes a reversible rearrangement that promotes long-lived lifetimes, a feature likely important for correct recognition of a 5′ SS. In this kinetic model, transitions into or out of the long-lived state are slow, and RNA dissociation from U1 occurs nearly 12-fold more rapidly than formation of the long-lived state (*Figure 1G*). We conclude that RNAs can be quickly released by U1 after association even if those RNAs contain a consensus 5′ SS as in RNA-4+2. Further, the presence of a consensus site does not result in long-lived complex formation occurring more quickly than dissociation.

Previous analysis of U1-binding events on immobilized RP51A pre-mRNAs in yeast WCE (yWCE) also resulted in multi-exponential dwell time distributions (*Hoskins et al., 2011*; *Larson and Hoskins, 2017*). The exponential fits of dwell times for RNA-4+2 binding to purified, immobilized U1 snRNP and for U1 snRNP (in WCE and without ATP) binding to immobilized RP51A pre-mRNAs containing the same 5' SS have similar parameters (*Larson and Hoskins, 2017*). In both cases, most binding events are short-lived and with lifetimes of ~12 s. The long-lived kinetic parameter was smaller (64 vs. 177 s) but with a larger amplitude (0.3 vs. 0.1) than the events we observed with purified U1. Longer binding events of ~200 s were observed in WCE with this 5' SS but only when either CBC or BBP were also capable of binding the pre-mRNA.

Together, our data indicate that short- and long-lived interactions with RNA substrates are an inherent property of U1. Since we purified and immobilized U1 and studied its interactions with small RNAs, the diversity of binding events cannot solely originate from the influence of *trans*-acting factors

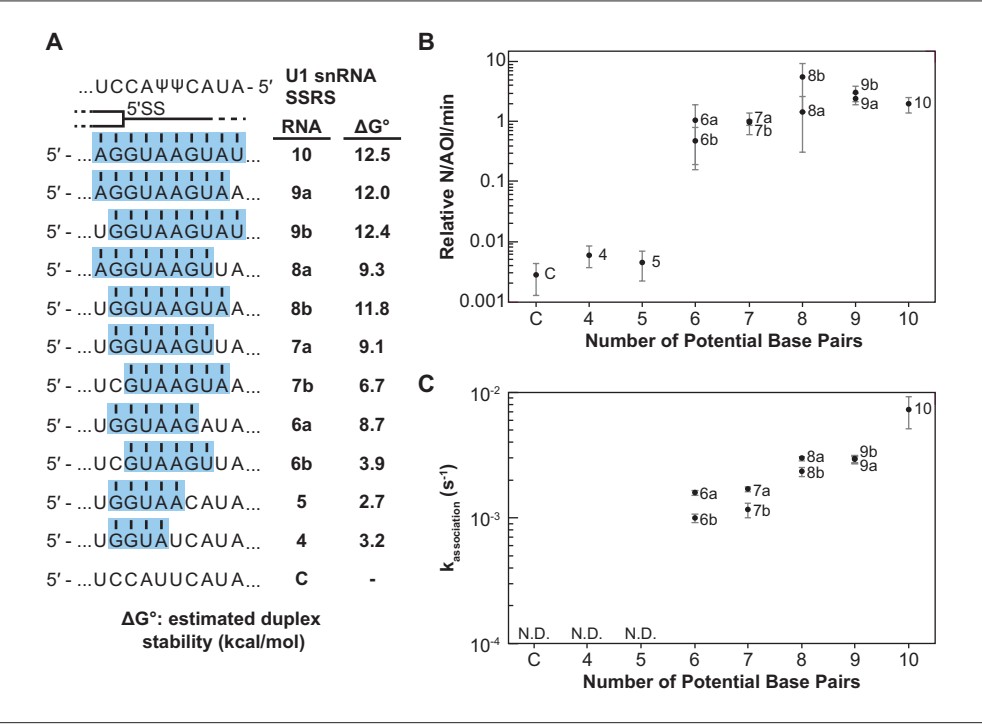

**Figure 2.** Impact of base-pairing potential on RNA oligo binding to U1. (**A**) RNA oligos tested for interaction with U1 containing 4–10 predicted base pairs and the calculated free energy changes for duplex unwinding/ formation based on nearest neighbor analysis. The regions shaded in blue are predicted to pair with the splice site recognition sequence (SSRS). (**B**) Relative event densities of oligo binding to immobilized U1 molecules as a function of potential base pairs. Ordinate values are computed as the number of binding events (**N**) per area of interest (AOI) per minute (min). (**C**) Measured association rates of the oligos to U1 as a function of potential base pairs. For (**B**), the plotted points represent the average results from at least three replicate experiments ± SD. For (**C**), the plotted points represent the fitted parameters ± the uncertainties of the fits. Numbers of events (*N*) are reported in *Figure 2—source data 1*.

The online version of this article includes the following source data for figure 2:

**Source data 1.** Number of measured events and calculated association rates for RNA oligos shown in *Figure 2*.

present in a WCE or folding/unfolding of large RNA substrates. We do not exactly know how these factors influence U1 binding in complex environments, but they may be the origins of differences we observed between experiments carried out with purified U1 and with U1 present in WCE.

## Base-pairing potential accelerates U1/RNA complex formation

We next systematically studied how the base-pairing potential of the RNA oligo influenced binding by U1 snRNP. We carried out single-molecule binding assays with RNAs capable of forming between 4 and 10 contiguous base pairs with the snRNA (*Figure 2A*). All these substrates can form base pairs at the highly conserved G+1 and U+2 positions of the 5′ SS, and we extended base pairing outward from these positions toward the 5' and 3' ends of the SSRS. For several positions we also varied the duplex position with pairing extending away from or toward the 5' end of the U1 snRNA without altering the number of potential base pairs (e.g., RNA-6a vs. -6b).

When the number of binding interactions to immobilized U1 and the apparent association rates were measured, the RNA oligos exhibited two observable and distinct classes of behavior. In the first class, RNA oligos capable of forming <6 contiguous base pairs showed very few colocalized binding events with U1 (*Figure 2B*). While these oligos may have been able to form sequence-specific interactions with U1, these interactions were either too rapid or infrequent for us to observe. The few measurable events were essentially indistinguishable in frequency to background binding of RNA-C. RNAs capable of forming ≥6 contiguous base pairs exhibited a second class of behavior. These RNAs had a 100-fold increase in detectable U1-binding event density compared to RNAs in the first class (*Figure 2B*). The dependence of the event density on the number of potential base pairs with the snRNA supports that the interactions are not only sequence dependent (*Figure 1*) but are also due to interactions with the U1 SSRS.

For RNAs with detectable U1 binding, we were able to calculate the observed association rate ($k_{association}$) to U1 under these conditions (*Figure 2C*, *Figure 2—source data 1*). RNAs capable of forming more potential base pairs with U1 bound more quickly. The correlation of the association rate with extent of base pairing could be due to RNAs with greater complementarity also having a greater probability of nucleating duplex formation due to the increased number of possible toeholds or short stretches of pairing interactions. This hypothesis is consistent with previous single-molecule fluorescence resonance energy transfer studies and ensemble measurements of DNA and RNA oligo hybridization that show nucleation of nucleic acid duplex formation by base-pairing interactions of only 2–4 nt in length (*Cisse et al., 2012*; *Craig et al., 1971*; *Marimuthu and Chakrabarti, 2014*; *Wetmur, 1991*; *Wetmur and Davidson, 1968*).

Additionally, we observed that oligos capable of pairing toward the 3' end of the SSRS formed observable complexes more quickly than those where the pairing was shifted toward the 5' end (*Figure 2C*, RNAs-6a, -7a, and -8a vs. -6b, -7b, and -8b). This indicates that the 3' end of the SSRS might be either more accessible to the RNAs or can more easily facilitate nucleation of RNA interactions that lead to the observable binding events. This latter possibility may be related to the increased calculated thermodynamic stability of duplexes with pairing interactions closer to the 3' end of the SSRS due to the presence of a G/C pair in this region: RNAs-6a and -7a are predicted to form more stable duplexes than RNAs-6b and -7b (*Figure 2A*).

## The abundance of short- and long-lived U1/RNA complexes depends on base pairing

We next studied the dwell times with U1 for the same series of RNA oligos. By visually inspecting the individual fluorescence time trajectories, we were immediately struck by apparent differences in binding behaviors. We frequently observed very short dwell times with RNAs capable of only forming a small number of base pairs (**RNA-6a**) and a mixture of short and long dwell times for RNAs capable of forming increasing numbers of base pairs (**RNA-8a and RNA-10**). When the individual dwell times from each experiment were combined and fit to single- or double-exponential functions, resulting probability density plots and kinetic parameters confirmed these observations (*Figure 3B*). RNA-6a is capable of only forming six base pairs with U1 and its distribution of dwell times could also be fit using only a single exponential ($\tau_S \approx 12$ s), consistent with short-lived binding. RNA-8a can make up to eight base pairs with U1 and its dwell times were best fit using an equation with two kinetic parameters describing short- ($\tau_S \approx 43$ s) and long-lived binding events ($\tau_L \approx 137$ s). RNA-10 also require two

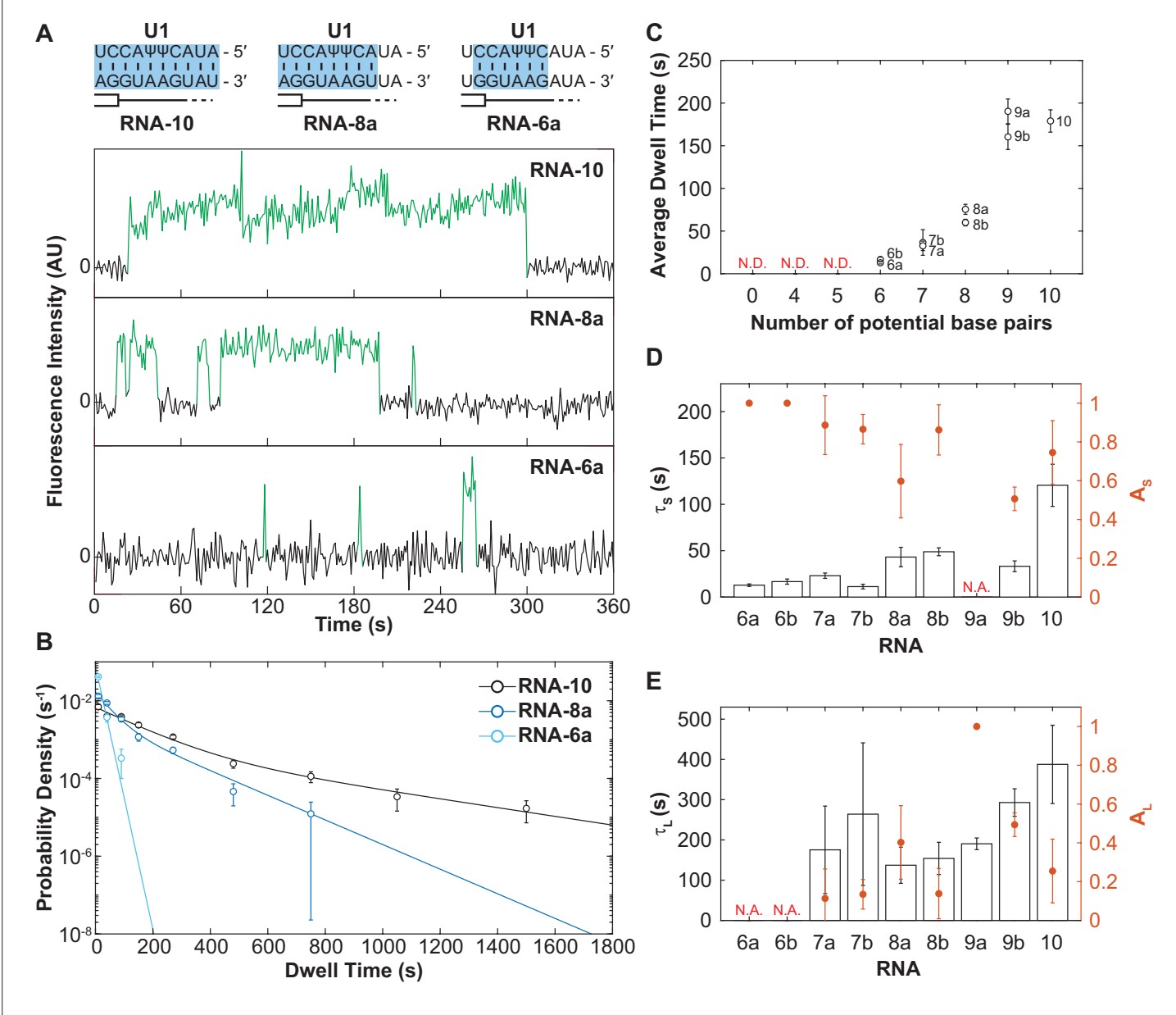

**Figure 3.** The long-lived state is dependent on the length of the small nuclear RNA (snRNA)-RNA duplex. (**A**) Representative fluorescence trajectories of changes in Cy3 intensity (green) due to oligo binding to a single immobilized U1 molecules for RNAs-6a, -8a, and -10. Also shown are the predicted pairing interactions (blue) between the oligos and the U1 SSRS. (**B**) Probability density histograms for dwell times for RNAs-6a, -8a, and -10 binding to U1. Lines represent the single- or double-exponential distribution obtained for the fitted parameter from each data set. (**C**) The average dwell time of each RNA oligomer in *Figure 2A*. The average dwell time is not determined (ND) for oligomers for which little binding was observed. (**D–E**) Bars shown the estimated parameters for short-lived binding (panel D, $\tau_S$ < 120 s) and long-lived binding (panel E, $\tau_L$ > 120 s) shown for each RNA oligomer in *Figure 2A* and correspond to the values on the left ordinate. If there is only one fit parameter, then the other is not applicable (NA). Orange markers show the amplitude of the time constant ($A_S$ and $A_L$) across the fitted distribution and correspond to the values on the right ordinate (orange). Error bars in C–E are standard error of the estimated parameters determined by bootstrapping. Numbers of events (*N*) and fit parameters are listed in *Figure 3— source data 1*.

The online version of this article includes the following source data for figure 3:

**Source data 1.** Number of measured events and calculated fit parameters for RNA oligos shown in *Figure 3*.

exponential parameters to describe the data, yielding longer 'short-lived' events ($\tau_S \approx 120$ s) as well as increased dwell times for the longer-lived events ($\tau_L \approx 355$ s).

When we examined all the RNAs in this series, we observed a trend: as the number of potential base pairs increased so did the average dwell time of the U1/RNA interaction (*Figure 3C*). RNAs that could only form a few potential base pairs possessed predominantly short-lived dwell times (defined here as $\tau_S < 120$ s) with a small fraction of long-lived ($\tau_L > 120$ s) binding events and correspondingly small amplitude for the long-lived kinetic parameter (*Figure 3D and E*). As the number of potential base pairs increased, generally so did the amplitude of $\tau_L$. It is unlikely that these results arose from presence of two subpopulations of U1 snRNPs in our experiments (one capable of only making short-lived interactions and one capable of only making long-lived interactions) since we would not expect these subpopulations to change in abundance between experiments carried out with the same preparations of U1. Furthermore, hidden Markov modeling of RNA-4+2 (which considers the relationship between consecutive binding and unbinding events) favored a sequential scheme involving RNA binding followed by a transition to long-lived state (*Figure 1G*), likely due to a conformational change of the U1 snRNP, rather than direct formation of two different bound state complexes.

Instead, these data are most consistent with a mechanism in which U1 association with RNAs involves multiple steps. All RNAs that we can observe interacting with U1 (those capable of making $\geq$6 base pairs) can form the short-lived complex. RNAs with a limited number of base pairs (i.e., RNAs-6a, -6b) rarely progress through the second step to form the long-lived complex and most often dissociate from the intermediate state. On the other hand, RNAs with many base pairs (RNAs-7–10) are more probable to transition to the long-lived complex.

Finally, it is interesting to note that RNAs in which the base pairing extends to the 3' end of the SSRS (*Figure 3C*, RNAs-8a and -9a) also had a larger average bound time than those capable of forming the same number of base pairs but not reaching the 3' end of the SSRS (RNAs-8b and -9b). This suggests that pairing within the 3'-most nt of the SSRS closest to the zinc finger of Yhc1 is not only important for increasing the rate of U1 binding but also contributes to formation of the longest-lived U1/RNA complexes. Combined, these results support formation of a short-lived, intermediate between U1 and RNAs that is dependent on base pairing for its formation. The RNA can then dissociate from this intermediate or the U1/RNA complex can transition to more tightly bound state.

## Some U1/5' SS duplexes are destabilized in the U1 snRNP

In addition to varying amplitudes, the short- and long-lived time constants from the fits ($\tau_S$ and $\tau_L$) also varied (*Figure 3D and E*). The short-lived dwell time parameter ($\tau_S$) ranged from 12 to 120 s for RNA oligos capable of forming 6–10 contiguous, potential base pairs. The long-lived dwell time parameter ($\tau_L$) ranged from 137 to 388 s for RNA oligos capable of forming 7–10 base pairs. For both parameters, RNAs capable of forming more base pairs also tended to have longer dwell times. With the exception of $\tau_S$ for RNA-10, $\tau_S$ and $\tau_L$ parameters only varied within a range of two- to fourfold. This was surprising since a previous single-molecule fluorescence study of RNA oligo hybridization reported a 10-fold decrease in off-rate due to presence of one additional base pair (*Cisse et al., 2012*).

It is possible that protein components of the U1 snRNP, in addition to the SSRS/5' SS base-pairing interactions, contribute to the small range in $\tau_S$ and $\tau_L$ we determined. To test this, we constructed a RNA-only mimic of the U1 SSRS (*Figure 4A*). In this case, binding kinetics would only be influenced by the nucleic acid complexes being formed and not be influenced by snRNP proteins or structural constraints imposed by U1. Unlike U1 snRNP, the surface-immobilized mimic did not efficiently bind to the RNA oligos when they were present in solution at nM concentrations (the upper concentration limit of our single-molecule assay). So, we instead pre-annealed each oligo to the mimic and then measured its off-rate by monitoring disappearance of colocalized oligo fluorescence signals over time (*Figure 4A and B*).

For each of the RNAs, we were able to fit the dissociation data to equations containing a single exponential term (*Figure 4—source data 1*). This signifies that the RNAs are dissociating in a single observable step from the immobilized mimic and that dissociation was occurring from only a single type of RNA/mimic complex. The single-exponential kinetics are in contrast with results obtained for many of the same RNAs with U1 snRNP, for which multi-exponential kinetic equations were required to fit the dwell time data (RNA-7a,b; -8a,b; -9b, and -10). This was true for both a mimic that, like U1,

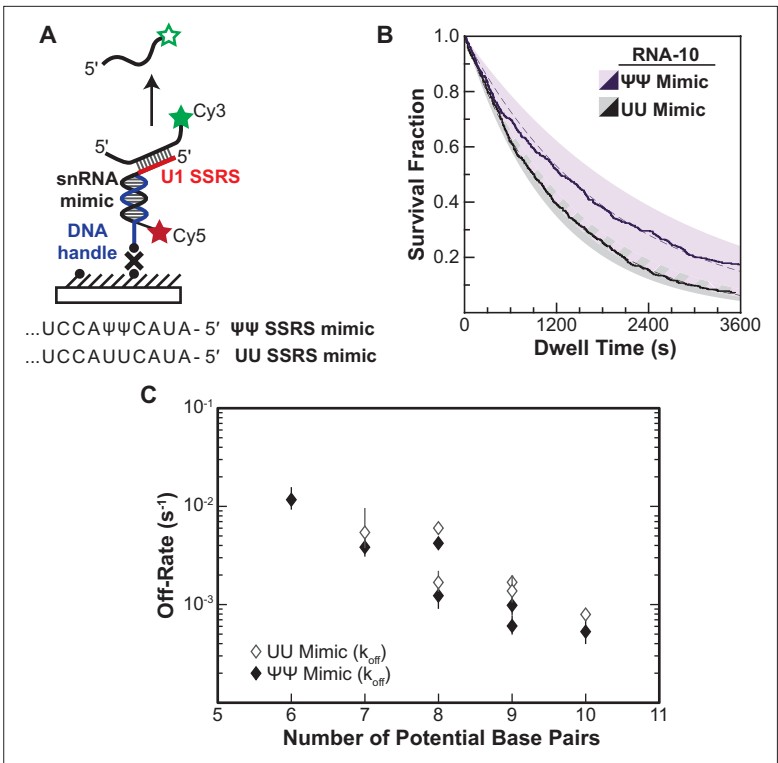

**Figure 4.** Lifetimes of 5' splice site (5' SS) oligo/RNA interactions are dependent on base-pairing potential in an RNA-only mimic of the U1 splice site recognition sequence (SSRS). (**A**). Schematic of a single-molecule assay for monitoring dissociation of RNA oligos from the RNA-only mimic of the U1 SSRS. Two mimics were used that contain pseudouridine (Ψ) or uridine (U) at two positions in the SSRS that have Ψ in the native U1 small nuclear RNA (snRNA). (**B**) The fraction of colocalized RNA oligos remaining was plotted over time to yield survival fraction curves for determining RNA oligo off-rates (black lines). The curves were then fit to exponential decay functions to yield off-rates as well as 95% confidence intervals for the fits (dashed lines and shaded regions, respectively). Shown are the survival fraction curves for RNA-10 dissociation (see **Figure 2A**). (**C**) Measured off-rates for RNA oligos to the SSRS mimics (see **Figure 4—source data 1** for rates and numbers of events, *N*) plotted as a function of potential base pairs.

The online version of this article includes the following source data for figure 4:

**Source data 1.** Number of measured events and calculated off rates for RNA oligos shown in **Figure 4**.

---

contains pseudouridines in the SSRS as well as for one with uridine substitutions at those positions. While a comparison of the mimic and snRNP data is limited by the different experimental conditions (pre-annealing vs. equilibrium binding), the measured data are consistent with non-identical dissociation pathways for a given RNA oligo between the RNA-only mimic and the U1 snRNP.

In addition to differences in the dissociation pathways, the amount of time the oligos remained bound differed dramatically between the RNA mimic and U1. Dissociation rates from the mimic varied linearly with base-pairing potential over 20-fold, a larger range than for binding of the same oligos to U1 snRNP (**Figure 4C**). Surprisingly, the lifetimes of many of the RNAs bound to the mimic were also much longer than their lifetimes bound to U1. For example, RNA-10 had a dissociation rate of $5.5\times10^{-4}$ $s^{-1}$ when bound to the pseudouridine-containing mimic. This corresponds to an average lifetime of 1818 s—more than fivefold larger than the $\tau_L$ obtained for binding of the same RNA to U1. Again, given the limitations of these experiments, the data are consistent with the possibility some U1/5' SS duplexes can be destabilized in the context of the U1 snRNP. Thus, the lifetimes of U1/5' SS interactions in the snRNP cannot be predicted from base-pairing potential or studies of model RNA duplexes alone.

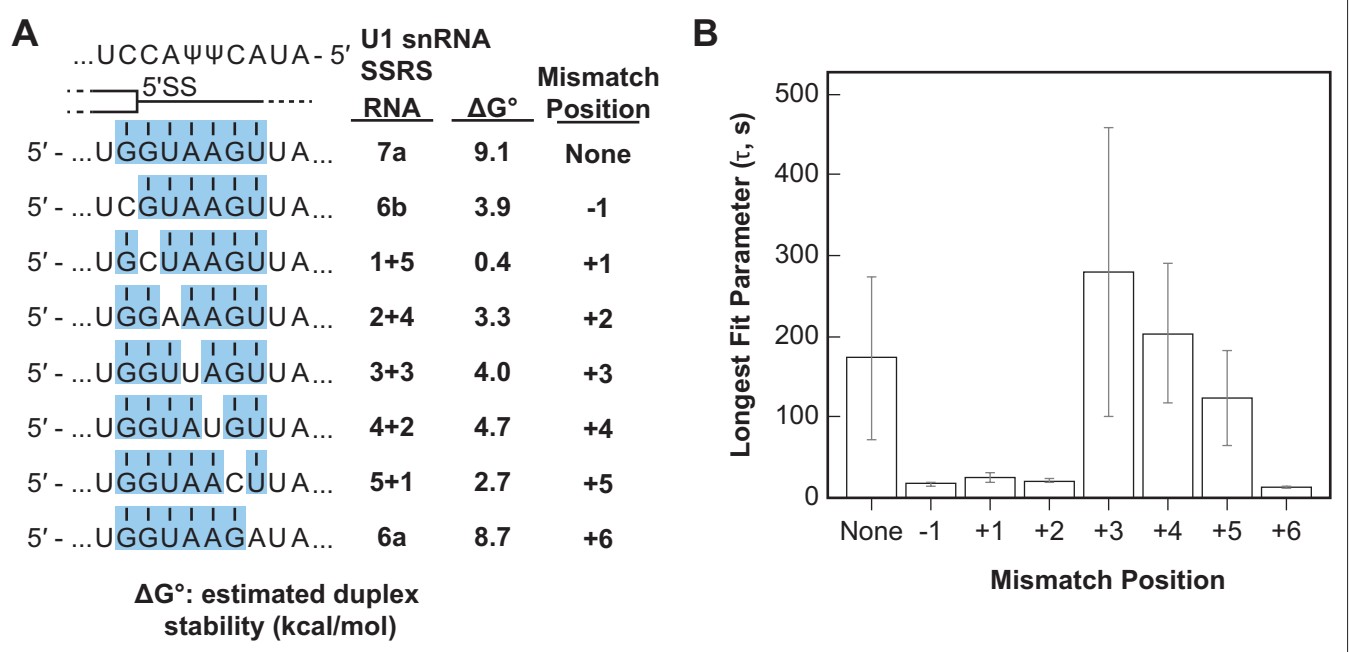

**Figure 5.** Long-lived U1/RNA interactions are dependent on mismatch position. (**A**) RNA oligos tested for interaction with U1 containing mismatches at the –1 to +6 positions and the calculated free energy changes for duplex unwinding/formation based on nearest neighbor analysis. The regions shaded in blue are predicted to pair with the splice site recognition sequence (SSRS). (**B**) The value of the longest-lived parameter ($\tau_0$ or $\tau_L$, see **Figure 5—source data 1** for fit parameters and numbers of events , $N$) obtained by fits of the distributions of dwell times to U1 for each RNA oligomer in panel (**A**). The plotted bars represent the fitted parameters ± the uncertainties of the fits. Note that data for RNA oligos 7a, 6a, and 6b were replotted from **Figure 3D and E** for comparison.

The online version of this article includes the following source data for figure 5:

**Source data 1.** Numbers of events and fit parameters for data shown in **Figure 5**.

## Long-lived U1/RNA interactions are sensitive to the location and type of mismatches

Splice sites with perfect and uninterrupted complementarity to U1 are very rare in yeast. In fact, only 14 annotated 5' SS in yeast contain six contiguous base pairs (corresponding to RNA-6b) and only one (a cryptic 5' SS in RPL18A) may contain more than seven contiguous base pairs (*Grate and Ares, 2002*). Most 5' SS are interrupted by one or more mismatches in complementarity to the U1 snRNA. We next tested how these mismatches impacted interactions of the RNA oligos with U1 snRNP. We analyzed and compared the binding interactions of RNAs capable of forming various numbers of contiguous base pairs between U1 SSRS nt +3 to +9. We incorporated mismatches systematically at each position resulting in RNAs that can form uninterrupted duplexes of seven or six base pairs (RNAs-7a, -6a, -6b) or interrupted duplexes of a total length of seven nucleotides (*Figure 5A*). One of the RNA oligos within this comparison group contains the U1 consensus 5' SS found within the well-spliced RP51A transcript (RNA-4+2, *Figure 5A*). Within this group, the mismatches result in a range of predicted duplex stabilities from –0.4 to 9.1 kcal/mol (*Figure 5*).

We observed long-lived complexes for RNA-7a, which has a 7 bp predicted duplex length, and only short-lived complexes for RNA-6a and -6b, which have only 6 bp predicted duplex lengths (*Figure 5B*; replotted from *Figure 3D and E*). Whether or not RNAs containing mismatches that disrupt the duplex with the SSRS showed long-lived interactions (like RNA-7a) or only short-lived interactions (like RNA-6a, -6b) depended on the position of the mismatch. Neither RNA oligos containing a C/C mismatch at the +1 site nor an A/A mismatch at the +2 site were able form long-lived complexes with U1. However, RNA oligos containing U/U mismatches at +3 or+4 or a C/C mismatch at +5 could form long-lived complexes (*Figure 5B*). From these data we conclude that long-lived complex formation is sensitive to the tested mismatches at some positions (+1, +2) and not others (+3, +4) within a substrate of 7 bp end-to-end length. In addition, the same type of mismatch (C/C) could either prevent or permit

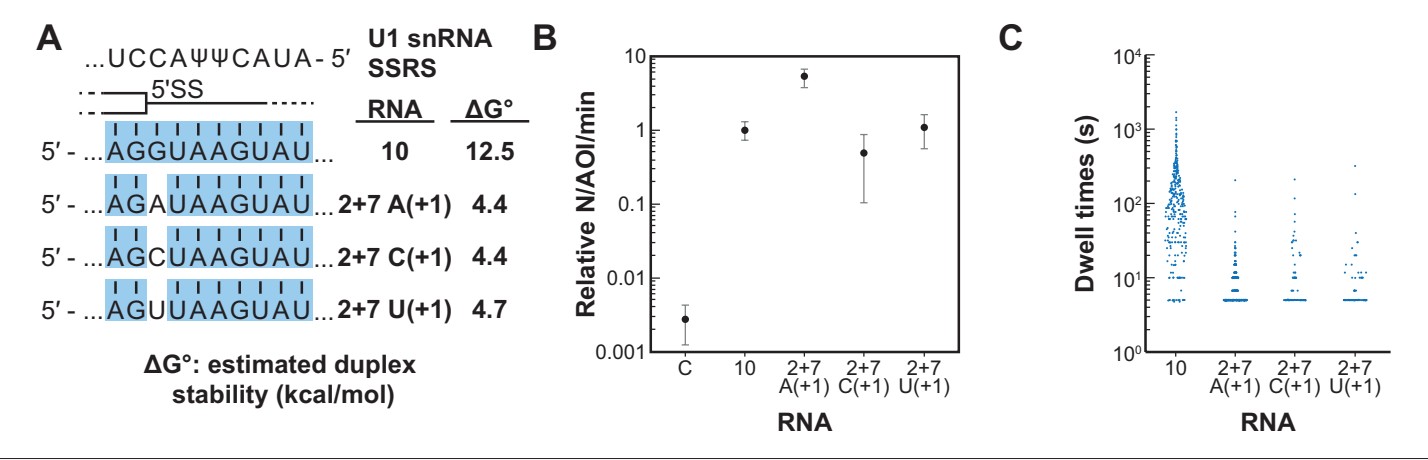

**Figure 6.** Long-lived interactions are greatly stimulated by G at the 5' splice site (5' SS) +1 position. (**A**) RNA oligos tested for interaction with U1 containing mismatches only at the +1 positions and the calculated free energy changes for duplex unwinding/formation based on nearest neighbor analysis. The regions shaded in blue are predicted to pair with the splice site recognition sequence (SSRS). (**B**) Relative event densities of oligo binding to immobilized U1 molecules for RNAs shown in panel (**A**). Ordinate values are computed as the number of binding events (**N**) per area of interest (AOI) per minute (min). Plotted are averages from replicate experiments ± SD (dots and vertical lines). (**C**) Distribution of observed dwell times for U1 interactions with oligos from panel (**A**). Each dot corresponds to a single dwell time for N = 295, 518, 90, or 89 events for RNAs 10 and 2+7 variants A(+1), C(+1), and U(+1) , respectively.

The online version of this article includes the following source data for figure 6:

**Source data 1.** Fit parameters and log-likelihood results for RNA oligos shown in *Figure 6*.

long-lived complex formation depending on its position within the 7 bp duplex. Consequently, formation of long-lived U1/5' SS interactions does not correlate with predicted duplex stabilities (cf., RNA-5+1 vs. RNA-2+4, -6a, or -6b in *Figure 5B*).

## Long-lived U1/RNA interactions depend on base pairing at the G+1 position of the 5' SS

We next tested if a single mismatch could eliminate long-lived binding even if all other positions within the 5' SS oligo could potentially pair with SSRS. We incorporated single mismatches at the +1 position of RNA-10 (*Figure 6A*). This results in a mismatch at the first position of the highly conserved 5' SS GU. All RNAs containing a mismatch at +1 were able to associate with U1 at rates ~100-fold greater than background binding by RNA-C (*Figure 6B*). However, none of them were able to form appreciable amounts of the long-lived complex (*Figure 6C*). The observed distributions of dwell times for RNAs containing mismatches at +1 could still be best fit to two exponential distributions containing short- and long-lived parameters (*Figure 6—source data 1*). However, the amplitudes of the long-lived parameters were very small as expected from the scarcity of the long-lived events. Consistent with data shown in *Figure 5*, the predicted thermodynamic stabilities again did not correlate with observation of the long-lived complexes. For example, RNA-2+7 (A+1) containing an A/C mismatch at the +1 position is predicted to form a more stable duplex than RNA-5+1 (ΔG° –4.4 to –2.7 kcal/mol). Yet, the amplitude of the long-lived parameter for RNA-5+1 is ~×14 greater than that for RNA-2+7 (A+1). These results show that long-lived complex formation between U1 and the RNA oligos is intolerant of mismatches at the +1 position. Failure of U1 to accumulate on RNAs with mismatches at the +1 site is not due to lack of association. Rather, recognition of a mismatch at +1 involves a discrimination step occurring after association and mismatched RNAs are rapidly released.

## Discussion

By studying single molecules of yeast U1 snRNPs interacting with a diverse range of RNA oligos, our experiments have revealed the dynamics associated with the earliest step of 5' SS recognition. U1 can form both short- and long-lived, sequence-dependent complexes with RNAs (*Figures 1 and 3*). RNA binding is accelerated by increased numbers of potential base pairs as well as by their positioning

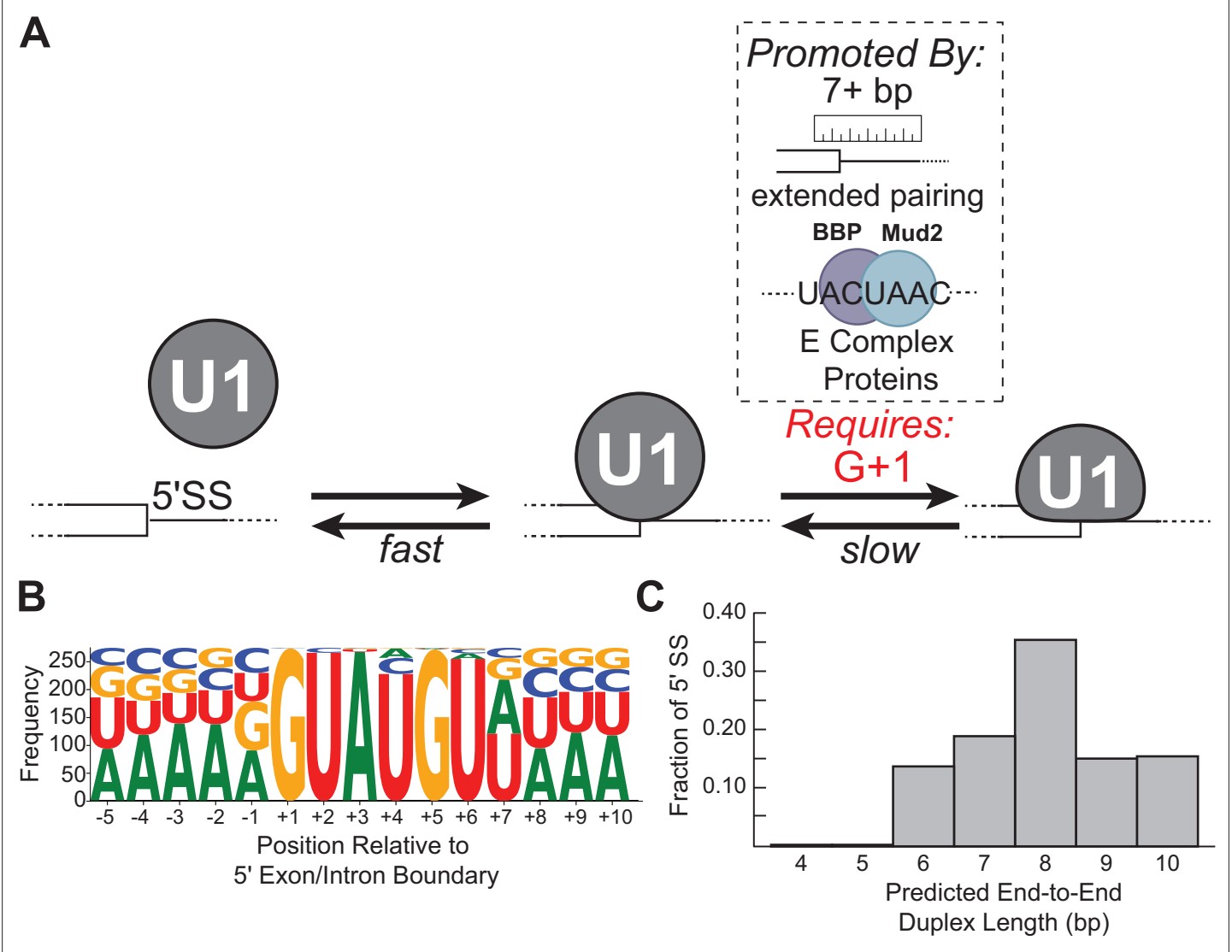

**Figure 7.** Multi-step authentication model for 5' splice site (5' SS) recognition. (**A**) U1 binding initially occurs by formation of a weakly interacting complex that is dependent on base-pairing potential between the RNA and U1 splice site recognition sequence (SSRS). Stable binding is dependent on presence of G+1 at the 5' SS and formation of an extended duplex with an end-to-end length of at least 7 bp or the presence of *trans*-acting splicing factors such as E complex proteins (***Larson and Hoskins, 2017***). (**B**) Sequence LOGO for annotated yeast 5' SS (***Lim and Burge, 2001***). (**C**) Histogram of end-to-end duplex lengths based on base-pairing potential between annotated yeast 5' SS (*N*=282) and the U1 SSRS. Most of these duplexes contain one or more mismatches with the SSRS.

closer to the 3' end of the SSRS—the same region in which the Yhc1 and Luc7 proteins contact the 5' SS/SSRS duplex (***Kondo et al., 2015***; ***Pomeranz Krummel et al., 2009***; ***Li et al., 2017***; ***Plaschka et al., 2018***; ***Figure 2***). Sequence-dependent interactions with lifetimes of several seconds are only observed with oligos capable of forming duplexes of at least 6 bp in length (***Figure 2***) while additional potential base pairing increases the probability of forming long-lived interactions lasting several minutes (***Figure 3***). Relative to an RNA-only mimic, U1 snRNP binds freely diffusing oligos more readily when they are present at nM concentrations and accelerates the release of RNAs capable of forming the largest number of potential base pairs (***Figures 3 and 4***), indicating that snRNP proteins and/ or U1 snRNA structure can destabilize the SSRS/5' SS duplex. Formation of long-lived interactions is dependent on the position of mismatches as well as pairing at the G+1 position rather than predicted thermodynamic stabilities of the RNA duplexes (***Figures 5 and 6***).

These results support a reversible multi-step binding process in which U1 first forms sequence-dependent short-lived interactions with RNAs via the SSRS and then transitions to a more stably

bound complex if certain requirements are met (*Figure 7A*). This agrees with previous single-molecule experiments with U1 in WCE (*Larson and Hoskins, 2017*) and biochemical studies of the temperature dependence of yeast U1/RNA interactions (*Du et al., 2004*) while providing further details of the 5' SS recognition process in the absence of *trans*-acting factors or confounding effects due to RNA secondary structures. We have not yet identified what specific event or conformational change is associated with long-lived complex formation; however, comparison of cryo-EM densities for free yeast U1 and the U1/U2-containing yeast pre-spliceosome (in which U1 is bound to a 5' SS) reveals that Yhc1 and Luc7 are more ordered in the latter (*Li et al., 2017*; *Plaschka et al., 2018*). It is possible that a disorder-to-ordered conformational change for Yhc1 and Luc7 results in the short- to long-lived transition inferred from our single-molecule assays. Mismatches at particular positions such as G+1 may inhibit this transition and/or prevent formation of stable contacts between the two proteins or with the RNA. In agreement with the importance of potential Yhc1/Luc7 contacts, mutants of Luc7 located at the Yhc1 interface exhibit numerous genetic interactions with U1 snRNP proteins and splicing factors involved in E complex formation (*Agarwal et al., 2016*; *Li et al., 2019*). We also note that interactions between U1 proteins and RNA sequences or structures not included in these experiments, such as RNA hairpins or binding sites for Nam8 (*Li et al., 2019*; *Plaschka et al., 2018*), may influence this process to further tune U1 binding on specific transcripts.

## Checkpoints during 5' SS recognition

Combined, our data also indicate that binding involves several checkpoints before long-lived complexes are formed (*Figure 7*). This scheme is consistent with 5' SS recognition involving conformational proofreading as has been implicated in many other nucleic acid recognition events including ribosome assembly and translation (*Rodgers and Woodson, 2019*; *Blanchard et al., 2004*). In the case of U1, the conformational change that leads to proofreading (rapid release of the RNA) or stable binding may involve rearrangement of Yhc1 and Luc7 as discussed in the preceding paragraph. Thus, the conformational proofreading would involve formation of different states of the U1 snRNP with different kinetic properties. The initial barrier to forming short-lived complexes is low and requires limited complementarity to the SSRS. Formation of this complex is readily reversible which permits rapid surveillance of transcripts by U1 for 5' SS and prevents accumulation of U1 on RNAs lacking features necessary for splicing.

Passage to the long-lived complex is more stringent and dependent on a G nucleotide at the +1 position as well as increased complementarity. It is likely that the need for pairing with G+1 evolved in U1 due to the importance of this nucleotide in splicing chemistry since G+1 at the 5' SS must form a non-Watson Crick pair with G-1 at the 3' SS during exon ligation (*Parker and Siliciano, 1993*; *Plaschka et al., 2019*; *Yan et al., 2019*). Thus, U1-binding kinetics prioritize identification of a nucleotide used for splicing chemistry even though U1 itself does not participate in those steps.

Interestingly, we found that formation of the long-lived state often correlated with predicted duplex length more so than predicted duplex stability (*Figure 5*, e.g., RNA-5+1 vs. RNA-6a or -6b). This suggests that 5' SS recognition involves a 'molecular ruler' like those observed in other RNPs (*Kwon et al., 2016*; *Macrae et al., 2006*). From our data, we would predict that U1 uses this ruler activity to preferentially form long-lived interactions with RNA duplexes >7 bp in end-to-end length. In terms of U1 snRNP structure, duplexes of 6–7 bp in length are needed to span the distance between the zinc finger of Yhc1 to the second zinc finger of Luc7 and the Yhc1/Luc7 interface (*Li et al., 2019*)—suggesting a role for duplex length in conformational changes or intermolecular contacts associated with stable 5' SS binding (*Figure 1B*).

A duplex length requirement of at least 7 bp is surprising since the consensus sequence of the 5' SS indicates only six highly conserved positions (*Figure 7B*). How then would most natural 5' SS in yeast be able to stably bind U1? When predicted end-to-end duplex lengths between the SSRS and yeast 5' SS are analyzed, rather than their specific nucleotide sequences, it is apparent that the overwhelming majority of 5' SS can form extended duplexes with U1 with the caveat that these duplexes may contain one or more internal mismatches (*Figure 7C*). Consequently, we would predict that most yeast 5' SS are able to meet duplex length requirements for long-lived complex formation.

An additional consequence of this length requirement and the duplexes described in *Figure 7C* is that they also favor base-pairing interactions between the 5' end of the U1 SSRS and the 3' end of the splice site. For example, each of the RNAs in our study capable of forming long-lived interactions also

could pair at the +6 position (G/GUAUG<u>U</u>) of the 5' SS. This particular position of the 5' SS is important since it also pairs with the '<u>A</u>CAGA' sequence of the U6 snRNA (base-pairing position underlined) to promote splicing catalysis (*Sontheimer and Steitz, 1993*; *Kandels-Lewis and Séraphin, 1993*; *Kim and Abelson, 1996*). As mentioned above for recognition of G+1, the kinetic properties of U1 are, in part, optimized to facilitate interactions between the 5' SS and the splicing machinery that are important for catalysis even after U1 is released.

Together these observations also suggest that for many consensus SS, U1 can recognize and be retained on introns even if other intron features, such as the branch site, have not yet been transcribed in agreement with models for fast, co-transcriptional recruitment of U1 (*Oesterreich et al., 2016*; *Tardiff et al., 2006*). However, on weak SS it is likely that non-U1 splicing factors play a role in bypass of the molecular ruler to permit U1 accumulation. As an example of bypass, we previously observed that pre-mRNAs containing a weak 5' SS with only a 5 bp end-to-end length can form long-lived complexes with U1 in WCE but only when either the CBC or BBP/Mud2 could also bind the pre-mRNA (*Larson and Hoskins, 2017*). One limitation of our model is that we do not yet know if these *trans*-acting factors can also bypass the need for pairing with G+1 in the assays used here. Regardless, it is notable that only the branch site-binding factors, BBP/Mud2, increased the probability of long-lived complex formation and not just its lifetime (*Larson and Hoskins, 2017*). This is consistent with the idea that retention of U1 on transcripts is governed through direct and indirect interactions with intronic sequences involved in splicing chemistry including both the 5' SS and branch site.

## Base-pairing potential does not predict U1 interaction kinetics

Our data show that the lifetime of the U1/5' SS interaction cannot be predicted based on the thermodynamic stability of the base-pairing interactions alone likely due to the influence of snRNP proteins. Rather, strong positional effects lead to prioritization of base pairing at particular positions for stable retention of U1 (*Figures 5 and 6*). How kinetic stability of U1 at a particular 5' SS correlates with its subsequent use by the spliceosome has not yet been determined. If 5' SS usage does correlate with U1 lifetimes, then our results would be in strongest agreement with computational models for 5' SS identification that consider positional effects and interdependencies rather than simply the thermodynamic stabilities of the predicted base pairs (*Roca et al., 2013*; *Yeo and Burge, 2004*).

Since failure to release U1 can inhibit splicing (*Staley and Guthrie, 1999*), we would expect that optimal promotion of splicing by yeast U1 would occur by balancing its recruitment and retention with release. This 'Goldilocks' model has previously been proposed for 5' SS recognition by human U1 (*Chiou et al., 2013*). Interestingly, we observed that the lifetimes of the longest-lived U1/RNA complexes were similar to one another (*Figures 3 and 5*) and lifetimes of the RNAs capable of making 9 or 10 base pairs were much shorter when bound to U1 than expected based on their off-rates from an RNA mimic of the SSRS given the limitations of our assay (*Figures 3 and 4*). U1 may equilibrate its interactions with 5' SS by both stabilizing and destabilizing RNA duplexes. As a result, U1's interactions with sequence-diverse substrates may all be 'just right' for subsequent steps in splicing. This, in turn, may impact the ATP requirement for U1 release during activation (*Staley and Guthrie, 1999*).

## Implications for human 5' SS recognition

While much of the catalytic machinery of the spliceosome is well conserved between yeast and humans, we do not yet know if the mechanism of 5' SS recognition we propose also holds true for human U1. Chemical probing data of human U1 revealed allosteric modulation of the SSRS based on positioning of splicing regulatory elements (*Shenasa et al., 2020*). Differing conformations of the SSRS could lead to differences in binding behavior and give rise to short- or long-lived binding interactions like those we observe with yeast U1. Whether or not the SSRS of yeast U1 displays similar changes in conformation has not been determined. Ensemble binding data for human U1/RNA interactions also revealed a strong preference for formation of stable complexes on pairing at the G+1 site as well as position- and mismatch-dependent effects at other locations (*Kondo et al., 2015*; *Tatei et al., 1987*). This agrees with our single-molecule data on yeast U1 interactions. While it is possible that yeast and human U1 may have differing pathways for splice site recognition, the outcome of longer-lived binding on particular RNA sequences may be the same.

Finally, the large number of non-obligate accessory factors that associate with human U1 may yield highly malleable pathways for RNA binding. Different factors may tune 5' SS recognition by a holo-U1

complex to yield distinct kinetic mechanisms. This in turn could lead to enhancement or repression of U1 accumulation at particular sites or functional differences between U1 complexes involved in splicing or telescripting (*Kaida et al., 2010*; *Oh et al., 2017*). The mechanism we propose for yeast U1 may be most relevant for the subset of human U1 snRNPs that associate with alternative splicing factors such as LUC7L or PRPF39 that are homologs of obligate components of the yeast U1 snRNP (*Li et al., 2017*).

### General features of nucleic acid recognition by RNPs

Other cellular RNPs face similar challenges as U1 in finding specific nucleotide sequences and preventing accumulation on close cognates. While Cas9 (involved in bacterial CRISPR-based immunity), Hfq (involved in bacterial small RNA regulation) and Argonaute (AGO, involved in mRNA repression and silencing) RNPs are involved in very different biological processes than RNA splicing, single-molecule studies of each of these RNPs reveal striking similarities with yeast U1 (*Globyte et al., 2019*; *Małecka and Woodson, 2021*; *Salomon et al., 2015*; *Sternberg et al., 2014*). All these RNPs exhibit kinetic behaviors that lead to prioritization of certain sequences over others and are distinct from 'all-or-nothing' models for hybridization of nucleic acids in the absence of proteins (*Cisse et al., 2012*; *Wetmur, 1991*; *Wetmur and Davidson, 1968*). In the cases of AGO and Cas9, correct base pairing with the micro-RNA seed sequence (AGO) or PAM (Cas9) is necessary for fast and stable binding. Rapid reversibility of this interaction ensures that these RNPs can dissociate and find other targets if mismatches are detected within the priority region. Additional base pairing with the target then leads to the most stable binding interaction, consistent with binding occurring in multiple steps. These results are analogous to reversible interrogation of RNAs by U1 that prioritizes pairing at the G+1 site and formation of extended duplexes for stable interaction. Cas9 also accelerates target search by diffusion along DNA molecules (*Globyte et al., 2019*; *Sternberg et al., 2014*). While this has not been directly tested with U1, tethering of U1 to the pol II transcription machinery (*Kotovic et al., 2003*; *Zhang et al., 2021*), rather than RNAs themselves, may lead to similar acceleration in binding site identification. Indeed, our kinetic modeling of U1 interactions with a consensus 5' SS containing RNA shows that RNA association is relatively slow and on the order of ~$10^5$ M$^{-1}$ s$^{-1}$. Association of U1 with the transcription machinery may be needed to increase the effective local concentration of substrate RNAs and to explain in vivo observations of fast, co-transcriptional binding.

### Methods

**Key resources table**

| Reagent type (species) or resource | Designation | Source or reference | Identifiers | Additional information |
|---|---|---|---|---|
| Strain, strain background (*Saccharomyces cerevisiae*) | BJ2168 (MATa prc1–407 prb1–1122 pep4–3 leu2 trp1 ura3–52 gal2) | Bruce Goode Lab *Crawford et al., 2008* | yAAH0001 | |
| Strain, strain background (*Saccharomyces cerevisiae*) | U1-SNAP-TAP (BJ2168 +SNP1::SNP1-fSNAP-Hyg+SNU71::SNU71-TAP-URA) | This study | yAAH0393 | See Methods, Tap Tagging of Yeast U1 snRNP |
| Recombinant DNA reagent | Plasmid for in vitro transcription of RP51A (pBS117) | Michael Rosbash Lab *Séraphin and Rosbash, 1991* | pAAH0016 | |
| Sequence-based reagent | U1 cOligo (DNA) | Integrated DNA Technologies | JL-U1 5' complement | 5'-CTT AAG GTA AGT AT |
| Sequence-based reagent | U1 RT Oligo (DNA) | Integrated DNA Technologies | SRH15 | 5'-TCA GTA GGA CTT CTT GAT |
| Sequence-based reagent | U1 snRNA mimic (UU, RNA) | Integrated DNA Technologies | SRH21 | 5'-AUA CUU ACC UUA AGA UAU CAG AGG AGA UCA AGA AG /3Cy5Sp/ |
| Sequence-based reagent | U1 snRNA mimic (Ψ Ψ, RNA) | Integrated DNA Technologies | SRH36 | 5'-AUA C Ψ Ψ ACC UUA AGA UAU CAG AGG AGA UCA AGA AG /3Cy5Sp/ |

*Continued on next page*

*Continued*

| Reagent type (species) or resource | Designation | Source or reference | Identifiers | Additional information |
|---|---|---|---|---|
| Sequence-based reagent | Handle for U1 mimic (DNA) | Integrated DNA Technologies | SRH22 | 5'-/Biotin/ TCT CTT CTT GAT CTC CTC TGA TAT CTT A |
| Sequence-based reagent | RNA-Cy3 oligomers | Integrated DNA Technologies | | See *Figure 1—source data 1* |
| Commercial assay or kit | Criterion TGX Precast Gel (4–20%) | Bio-Rad | Cat. No. 567-1093 | |
| Commercial assay or kit | Silver Stain Plus Kit | Bio-Rad | Cat. No. 161-0449 | |
| Chemical compound, drug | GE Healthcare IgG Sepharose 6 Fast Flow resin | VWR Scientific | Cat. No. 95017-050 | |
| Chemical compound, drug | Calmodulin Affinity Resin | Agilent | Cat. No. 214303 | |
| Chemical compound, drug | Rnasin Ribonuclease Inhibitor | Promega | Cat. No. N2611 | |
| Chemical compound, drug | Pierce Protease Inhibitor Tablet | Thermo Fisher Scientific | Cat. No. A32965 | |
| Chemical compound, drug | TEV Protease | Sigma-Aldrich | Cat. No. T4455 | |
| Chemical compound, drug | BG-649-PEG-biotin | *Smith et al., 2013* | | |
| Chemical compound, drug | m7G(5')ppp(5')G RNA Cap Structure Analog | New England BioLabs | Cat. No. S1404S | |
| Chemical compound, drug | AMV Reverse Transcriptase | Promega | Cat. No. M5101 | |
| Chemical compound, drug | RnaseH (2 U/µL) | Thermo Fisher Scientific | Cat. No. 18021014 | |
| Chemical compound, drug | Vectabond | Thermo Fisher Scientific | Cat. No. NC9280699 | |
| Chemical compound, drug | Biotin-PEG-SVA (MW 5000) | Laysan Bio | Cat. No. Biotin-PEG-SVA-5000-100 mg | |
| Chemical compound, drug | mPEG-SVA (MW 5000) | Laysan Bio | Cat. No. mPEG-SVA-5000-1G | |
| Chemical compound, drug | Poly-L-lysine | Sigma-Aldrich | Cat. No. P7890 | |
| Chemical compound, drug | Glucose Oxidase from Aspergillus niger type VII | Sigma-Aldrich | Cat. No. G2133-50KU | |
| Chemical compound, drug | Catalase from bovine liver | Sigma-Aldrich | Cat. No. C40-100MG | |
| Chemical compound, drug | (±)–6-Hydroxy-2,5,7,8-tetramethylchromane-2-carboxylic acid (Trolox) | Sigma-Aldrich | Cat. No. 238813-1G | |
| Chemical compound, drug | TransFluoSpheres Streptavidin-Labeled Microspheres (488/645), 0.04 µm, 0.5% solids | Life Technologies/ Invitrogen | Cat. No. T-10711 | |
| Chemical compound, drug | Yeast tRNA (10 mg/mL) | Thermo Fisher Scientific | Cat. No. AM7119 | |
| Chemical compound, drug | Streptavidin, 10 mg | Prozyme | Cat. No. SA10-10mg | |
| Chemical compound, drug | Heparin sodium salt from porcine intestinal mucosa | Sigma-Aldrich | H4784-250MG | |

*Continued on next page*

*Continued*

| Reagent type (species) or resource | Designation | Source or reference | Identifiers | Additional information |
|---|---|---|---|---|
| Chemical compound, drug | MilliporeSigma Calbiochem BSA, 10% Aqueous Solution, Nuclease-Free | Thermo Fisher Scientific | Cat. No. 12-661-525ML | |
| Software, algorithm | ImageQuant TL 8.1 software | GE Healthcare Life Sciences | https://www.gelifesciences.com | |
| Software, algorithm | MATLAB | MathWorks | https://www.mathworks.com/products/matlab.html | |
| Software, algorithm | ChemDraw Prime 15.0 | PerkinElmer | http://www.cambridgesoft.com/ | |
| Software, algorithm | Imscroll | *Friedman and Gelles, 2015* | https://github.com/gelles-brandeis/CoSMoS_Analysis | |
| Software, algorithm | QuB | *Nicolai and Sachs, 2013* | https://qub.mandelics.com | |
| Software, algorithm | DISC | *White et al., 2020* | https://github.com/ChandaLab/DISC | |
| Other | Ultra-clear centrifuge tubes (14 mL capacity) | Beckman Coulter | Cat. No. 344060 | Ultracentrifuge tubes for preparing yeast splicing extract |
| Other | Precision Plus Protein All Blue Prestained Protein Standards | Bio-Rad | Cat. No. 161-0373 | Protein molecular weight ladder for SDS-PAGE |
| Other | 0.8×4 cm Poly-Prep Chromatography Columns | Bio-Rad | Cat. No. 731-1550 | Columns used for TAP purification |
| Other | 10 kDa MWCO Slide-A-Lyzer dialysis cassette | Thermo Fisher Scientific | Cat. No. 66380 | Dialysis membranes used during purification |
| Other | Amicon Ultra 100 kDa MWCO centrifugal filters | Sigma-Aldrich | Cat. No. Z677906-24 | Concentrators used during purification |
| Other | Gold Seal Cover Slips (#1, 24×60 mm) | Thermo Fisher Scientific | Cat. No. 5031132 | Glass slides used in CoSMoS assays |
| Other | Gold Seal Cover Slips (#1, 25×25 mm) | Thermo Fisher Scientific | Cat. No. 3307 | Glass slides used in CoSMoS assays |
| Other | Fisherbrand Five-Slide Mailer | Thermo Fisher Scientific | Cat. No. HS15986 | Slide holder used to clean slides |

## TAP tagging of yeast U1 snRNP

C-terminal TAP and fSNAP tags were appended to the endogenous SNU71 and SNP1 proteins, respectively, by homologous recombination in the protease-deficient *S. cerevisiae* strain BJ2168 and selection for growth in the absence of uracil (TAP) or in the presence of hygromycin (fSNAP) (*Larson and Hoskins, 2017*; *Puig et al., 2001*).

## Purification of labeled U1 snRNP

A total of 10 L U1-SNAP-TAP yeast cultures were grown in 1 L batches of rich media (YPD) in a shaking incubator (30°C, 220 rpm) to late log stage. The cells were pelleted, washed, and resuspended in 3.5 mL (per 1 L culture) Lysis Buffer (10 mM Tris-Cl pH 8.0, 300 mM NaCl, 10 mM KCl, 0.2 mM EDTA, 5 mM imidazole, 10% v/v glycerol, 0.1% v/v NP40, 1 mM PMSF, 0.5 mM DTT). The resuspended cells were flash frozen in a drop-wise fashion in liquid nitrogen and stored at –80°C until lysed. The frozen pellets were lysed in batches using a Retsch Mixer Mill MM 400 (five rounds of 3 min at 10 Hz, with 2 min cooling in liquid nitrogen between rounds). The frozen lysate powder was stored at –80°C.

The total cell lysate from 10 L was thawed at 4°C. Lysis Buffer (10 mL) was used to dissolve one EDTA-free Protease Inhibitor Tablet (Pierce), and this solution was combined with the cell lysate. Insoluble material was removed by centrifugation (15,000 rpm, 4°C, 30 min). The supernatant was then cleared in an ultracentrifuge at 36,000 rpm, 4°C, for 75 min. The resulting middle layer was carefully

removed and added to 300 µL GE Healthcare IgG Sepharose 6 Fast Flow resin that had been equilibrated with IgG150 Buffer (10 mM Tris-Cl pH 8.0, 150 mM NaCl, 10 mM KCl, 1 mM MgCl$_2$, 5 mM imidazole, 0.1% v/v NP40, no reducing agent) to incubate at 4°C with rotation for 2 hr.

The resin slurry was divided between two 0.8×4 cm Poly-Prep Chromatography Columns. After the lysate had flowed through and without the resin running dry, each column was washed with 3×10 mL IgG150 Buffer (plus 1 mM DTT) containing one dissolved Protease Inhibitor Tablet (Pierce) per 50 mL buffer. The flow was stopped by capping the columns with 1.0 mL of resin plus solution remaining. TEV protease (40 U) and the SNAP-tag dye (2 µM) were then added. The columns were sealed with caps, and the resin was incubated for 45 min at room temperature in the dark with mixing. Subsequent steps were carried out with as little exposure to light as possible.

The labeled, TEV-cleaved eluent was added directly to calmodulin affinity resin (400 µL) that had been equilibrated with Calmodulin Binding Buffer (10 mM Tris-Cl pH 8.0, 150 mM NaCl, 10 mM KCl, 1 mM MgCl$_2$, 5 mM imidazole, 2 mM CaCl$_2$, 0.1% v/v NP40, no reducing agent, 4°C). The IgG resin was washed with an additional 200–300 µL IgG150 Buffer (plus 1 mM DTT) to ensure all sample was transferred to the calmodulin resin. To this slurry, three equivalent volumes (with respect to the volume of the TEV eluate) of Calmodulin Binding Buffer containing 10 mM β-mercaptoethanol were added. The slurry was then incubated at 4°C with rotation for 60 min.

The resin slurry was divided between two 0.8×4 cm Poly-Prep Chromatography Columns. After the flow through was eluted, each column was washed with 3×5 mL Calmodulin Binding Buffer containing 10 mM β-mercaptoethanol. Before elution, the columns were capped at the bottom to control the timing of subsequent steps. Buffer exchange was performed by washing the resin with 100 µL (approximate resin bed volume) Calmodulin Elution Buffer (10 mM Tris-Cl pH 8.0, 150 mM NaCl, 10 mM KCl, 1 mM MgCl$_2$, 5 mM imidazole, 4 mM EGTA, 0.08% v/v NP40, 10 mM β-mercaptoethanol). Immediately afterward, labeled U1 snRNP was eluted in 4×150 µL fractions using incubation times of 0, 2.5, 5, and 10 min with the elution buffer.

Fractions were then analyzed by SDS-PAGE, and fractions E1-E3 typically had the highest concentrations of U1. These fractions were pooled and dialyzed in a 10 kDa MWCO Slide-A-Lyzer dialysis cassette in 1 L Dialysis Buffer (10 mM Tris-Cl pH 8.0, 150 mM NaCl, 10 mM KCl, 1 mM MgCl$_2$, 5 mM imidazole, 10 mM β-mercaptoethanol) overnight at 4°C. In the morning, the cassette was moved to fresh Dialysis Buffer (1 L) for 4 hr. The dialyzed sample was concentrated in an Amicon Ultra 100 kDa MWCO centrifugal filter unit (14,000 rpm, 4°C, in 1 min intervals). The sample was mixed by pipetting up and down between spins and by addition of more dialyzed sample. The final sample volume (~100 µL) was divided into 5 µL aliquots, flash frozen, and stored at –80°C.

## HPLC-ESI-MS/MS analysis

Samples were analyzed by HPLC-ESI-MS/MS using a system consisting of a high-performance liquid chromatograph (nanoAcquity, Waters) connected to an electrospray ionization (ESI) Orbitrap mass spectrometer (LTQ Velos, Thermo Fisher Scientific). HPLC separation employed a 100×365 mm fused silica capillary micro-column packed with 20 cm of 1.7-µm-diameter, 130 Å pore size, C18 beads (Waters BEH), with an emitter tip pulled to approximately 1 µm using a laser puller (Sutter Instruments). Peptides were loaded on-column at a flow rate of 400 nL/min for 30 min and then eluted over 120 min at a flow rate of 300 nL/min with a gradient of 2–30% acetonitrile in 0.1% formic acid. Full-mass profile scans were performed in the orbitrap between 300 and 1500 m/z at a resolution of 60,000, followed by 10 MS/MS HCD scans of the 10 highest intensity parent ions at 42% relative collision energy and 7500 resolution, with a mass range starting at 100 m/z. Dynamic exclusion was enabled with a repeat count of two over the duration of 30 s and an exclusion window of 120 s.

## Activity assays

Splicing extracts (yWCE) were prepared from a BJ2168-derived strain of *S. cerevisiae* as previously described (*Ansari and Schwer, 1995*). Aliquots were flash frozen in liquid nitrogen, stored at –80°C, and thawed on ice once before use. Capped, [$^{32}$P] -labeled RP51A pre-mRNA was prepared by in vitro transcription and gel purified and splicing conditions were adapted from previously described protocols (*Crawford et al., 2008*). Splicing reactions contained 100 mM potassium phosphate pH 7.3, 3% w/v PEG-8000, 2.5 mM MgCl$_2$, 1 mM DTT, 2 mM ATP, 0.4 U/µL Rnasin, 40% v/v yWCE, 0.2 nM [$^{32}$P]-labeled RP51A pre-mRNA, and 0.048 U/µL RnaseH. To ablate the U1 snRNA, these reactions

were first prepared without ATP, Rnasin, [$^{32}$P]-labeled RP51A, or U1 snRNP and with the inclusion of 0.016 µg/µL U1 cOligo (5′-CTTAAGGTAAGTAT-3′) so that RnaseH would digest the 5′ end of endogenous U1 snRNA in the yWCE (*Du and Rosbash, 2001*; *Larson and Hoskins, 2017*). After 30 min at 30°C, the remaining components of the splicing reaction were added along with 0.04 µg/µL purified U1 snRNP. As controls, reactions were prepared without purified U1 snRNP or without U1 cOligo in the ablation reaction. After 60 min at room temperature the reactions were stopped, and RNA was extracted as previously described (*Crawford et al., 2008*). The products were resolved on a 9% acrylamide (19:1) gel (8 M urea, ×1 TBE buffer). The gel was dried and imaged using a Phosphor Screen and a Typhoon FLA 9000. The bands were quantified using ImageQuant software.

## 5' End analysis by dideoxynucleotide sequencing

RNA from purified U1 snRNP or 40 µL U1-SNAP-TAP yWCE was isolated by phenol-chloroform extraction and ethanol precipitation. All of the RNA isolated from labeled, purified U1 snRNP was used for reverse transcription while only 10% of the isolated RNA from yWCE was necessary. The isolated RNA was combined with 1 pmol [$^{32}$P]-labeled primer complementary to nucleotides 27–44 of U1 snRNA (5′- TCAGTAGGACTTCTTGAT) in Annealing Buffer (250 mM KCl, 10 mM Tris pH 8.0) and the reaction was incubated at 90°C for 3 min, snap cooled on ice for 3 min, then pre-heated to 45°C for 5 min. A reverse transcriptase (×2 RT) master mix was prepared containing 1 U/µL AMV Reverse Transcriptase in 25 mM Tris pH 8.0, 8 µM DTT, and 0.4 mM dNTPs.

For dideoxynulceotide sequencing, five parallel reactions were set up for each sample. The reactions were made with 3.0 µL ×2 RT master mix, 1.0 µL ddNTP/H$_2$O (1 mM ddATP, 1 mM ddCTP, 1 mM ddTTP, 0.3 mM ddGTP, or Rnase-free water), and 2.0 µL annealing reaction. The reverse transcription reaction was incubated at 45°C for 45 min. To stop the reaction, 2 µL formamide loading dye (95% v/v deionized formamide, 0.025% w/v bromophenol blue, 0.025% w/v xylene cyanol FF, 5 mM EDTA pH 8.0) was added then the samples were cooled on ice for 3 min then heated to 90°C for 3 min. A portion (5 µL) of each reaction was loaded onto a 0.4 mM thick 20% acrylamide (19:1) / 7.5 M urea/×1 TBE gel. The gel was run until bromophenol blue neared the bottom. The gel was dried and imaged using a phosphorscreen and a Typhoon FLA 9000.

RNA oligo secondary structure prediction and calculation of free energy of unwinding mFold was used to identify potential stable secondary structures formed by the RNA oligos (*Zuker, 2003*).

The approximate stability of the duplex between U1 snRNA or the U1 mimic RNAs and the Cy3-RNA oligomers was predicted by calculating the stability of hybridization of the uridine-substituted SSRS to the complementary sequence of the RNA oligo using the Hybridization function of DINAMelt (*Markham and Zuker, 2005*). We note that while base pairs with roloxridines are predicted to be more stable than those to uridine (*Deb et al., 2019*; also see *Figure 4*), thermodynamic parameters for base pairing to consecutive roloxridines bases, such as those found within the U1 SSRS (5′-AUAC Ψ ΨAC-CU-3′), have not to our knowledge been determined. Therefore, we were unable to use nearest-neighbor methods to calculate the thermodynamic stabilities for RNAs pairing to the U1 SSRS and instead approximated these stabilities using a uridine-substituted SSRS.

## Microscope slide preparation

Microscope slides and coverslips were cleaned and assembled into flow as previously described (*Crawford et al., 2008*). Briefly, top and bottom coverslips were cleaned by sonication for 60 min at 40°C in successive washes of 2% v/v Micro-90 solution, absolute ethanol, 1 M KOH and water with intermittent rinsing with MilliQ water between each wash step. The cleaned coverslips were silanized using freshly prepared 1% v/v Vectabond in acetone (~30 mL to cover) for 10 min at room temperature. After silanization, the slides were immediately and thoroughly rinsed with absolute ethanol. The coverslips were thoroughly dried again then assembled into flow cells using vacuum grease to demark lanes.

Poly-L-lysine-*graft*-PEG copolymer (PLL-*g*-PEG) passivation was used to coat the slide surface and heparin was included in slide washing and imaging buffers to produce a negatively charged surface (*Salomon et al., 2015*). Dry aliquots (2 mg) of PLL-*g*-PEG were dissolved to a final concentration of 4 mg/mL PLL-*g*-PEG in 100 mM HEPES-KOH pH 7.4 just before use. The silanized lanes of the flow cell were filled with the PLL-*g*-PEG solution (~30 µL each) and incubated at room temperature overnight in

the dark. For experiments using the U1 snRNA oligo mimic, slides were coated with PEG as previously described (*Crawford et al., 2008*).

## U1 snRNA mimic preparation

The U1 snRNA mimic, the biotinylated DNA handle, and an RNA oligomer, were annealed and the tripartite complex was immobilized on the slide surface. Annealing reactions consisted of 2 µM Cy5-labeled U1 mimic (UU or $\Psi\Psi$), 200 nM biotinylated DNA handle, and 10 µM Cy3-labeled RNA oligomer in 50 mM Tris-HCl pH 7.4, 400 mM NaCl. The reactions were heated to 95°C in a thermo-cycler and cooled by decreasing temperature in 5°C intervals every 2 min until the reaction reached 25°C. After heating, reactions were immediately stored on until use in single-molecule experiments.

## Single-molecule microscopy

CoSMoS experiments were performed on a custom-built, objective-based micromirror total internal reflection fluorescence microscope (*Larson et al., 2014*). The red laser (633 nm) was set to 250 µW, and the green laser (532 nm) was set to 400 µW for data collection. The fluorescence signal was imaged at 1 s exposure at 5 s intervals unless otherwise specified. For all experiments, the imaging buffer included glucose, glucose oxidase, and catalase, as oxygen scavengers (OSS), and rolox as a triplet state quencher (TSQ) (*Crawford et al., 2008*). Drift correction was performed, as necessary, by tracking the movement of individual immobilized spots for the duration of the experiment. Auto-focusing was carried out using a 785 nm laser and was done every minute in-between exposures. Mapping files were generated each day using TransFluorSpheres (Thermo Fisher Scientific) fluorescent in both the <635 and >635 nm fields of view (FOV).

For experiments with the U1 snRNA mimic, prepared slides were first washed with 200 µL Annealing Buffer (50 mM Tris-HCl pH 7.4, 400 mM NaCl) with 0.01 mg/mL yeast tRNA. Prior to imaging, each lane was washed with 70 µL 0.2 mg/mL streptavidin in Annealing Buffer (+tRNA) which was imme-diately washed away with 70 µL Annealing Buffer (+tRNA). The annealed mimic/handle/oligomer complex was diluted by a factor of 1:2000–5000 in Annealing Buffer (+OSS +TSQ +tRNA) and the lane was washed with 70 µL of this solution. The accumulation of the complex on the slide surface was monitored in real time in the >635 nm FOV and when the desired density of spots was achieved, excess components were washed away using 70 µL Annealing Buffer (+OSS +TSQ +tRNA). To initiate movies, the buffer in the lane was exchanged with 90 µL Mock Splicing Buffer (100 mM potassium phosphate pH 7.3, 10 mM HEPES-KOH pH 7.9, 20 mM KCl, 2.5 mM MgCl$_2$, 8% v/v glycerol, 5% w/v PEG-8000, 1.4 mM DTT, +OSS +TSQ +tRNA) and data recording was immediately started. For less stable complexes (e.g., RNA-7a), 30 min of data collection was sufficient to observe the dissociation of most (>90%) RNA oligomers. For the most stable complexes (e.g., $\Psi\Psi$ mimic + RNA-10), 80 min of data collection was necessary and imaging intervals were reduced to 1 exposure/10 s.

For experiments with U1 snRNP, prepared slides were first washed with 200 µL Mock Splicing Buffer with 0.05 mg/mL heparin and 0.01 mg/mL yeast tRNA. Subsequent steps were performed one lane at a time. The lane was washed with 70 µL 0.2–0.4 mg/mL streptavidin in Mock Splicing Buffer (+heparin +tRNA) which was allowed to incubate with the slide surface for 2–5 min. The lane was then washed with 70 µL 0.5 mg/mL (×10) heparin in Mock Splicing Buffer (+tRNA) and incubated for 10 min before U1 snRNP was added. U1 snRNP was diluted to a final concentration of 5–20 nM in Mock Splicing Buffer (+heparin +tRNA +OSS +TSQ) and added to the lane. The accumulation of fluorescent spots was monitored periodically in the >635 nm FOV until an optimal density was achieved (usually 2–5 min), then the excess complexes were washed away with 70 µL Mock Splicing Buffer (+heparin +tRNA +OSS +TSQ). Finally, the lane was washed with 70 µL 10 nM RNA-Cy3 in Mock Splicing Buffer (+heparin +tRNA +OSS +TSQ) and data recording was immedi-ately started. In U1 snRNP experiments, the Cy3 signal was typically imaged at 1 s exposure at 5 s intervals for 360 frames (30 min). We determined that the lifetimes measured in these experiments were not being limited by photobleaching by performing control experiments where the power of the 532 nm laser was varied from 200 to 600 µW or where the periodicity of the 1 s exposure was increased to 10 s.

Raw microscopy source data can be downloaded using Figshare at the link below: https://doi.org/10.6084/m9.figshare.c.6164067.

## Data analysis

Data analysis was performed as previously described (*Hoskins et al., 2011*; *Shcherbakova et al., 2013*). In brief, the fluorescence signal detected in the >635 FOV was used to select areas of interest (AOIs). After drift correction, these locations were mapped to the <635 FOV and the pixel intensity was integrated for each AOI using custom MATLAB software (*Friedman and Gelles, 2015*). Each colocalization event was manually inspected to confirm the presence of a colocalized spot in the AOI.

For fitting dwell times of oligos binding to the U1 mimic, the distributions were analyzed using survival fraction plots and fit with single-exponential decay functions which generated a 95% confidence interval (CI) for the calculated $k_{off}$ and an R-square parameter for the fit. The reciprocal of $k_{off}$ is the mean lifetime ($\mu$).

For analysis of oligo binding to immobilized U1 snRNPs, the distribution of observed dwell times was visualized as a probability density plots. To construct these plots, the dwell times were binned, and the probability of each bin was divided by the product of the bin width and the total number of events in the data set to compute a probability density value. The ordinate values are plotted on the log-scale to clearly show the difference in fitted time constants. Bin values were chosen to adequately represent the underlying distribution. Error bars for each bin were calculated as previously described based on the error of binomial distributions (*Hoskins et al., 2011*). These plots are overlaid with maximum likelihood estimates of a single- or double-exponential distribution as described by *Equations 1 and 2*, respectively (*Hoskins et al., 2011*). In these equations, $t_m$ is the time between consecutive frames and $t_{max}$ is the duration of the experiment; $A_1$ and $A_2$ are the fitted amplitudes for a bi-exponential distribution; and the 'taus' are the fitted dwell time parameters for single- ($\tau_0$) or bi-exponential ($\tau_1$ and $\tau_2$) distributions. Errors in the fit were determined by bootstrapping 1000 random samples of the data and determining the standard deviation of the resulting normal distribution.

$$\left[ \left( A_0 \cdot \left( e^{-\frac{t_m}{\tau_0}} - e^{-\frac{t_{max}}{\tau_0}} \right) \right) \right]^{-1} \cdot \left[ \frac{A_0}{\tau_0} \cdot e^{-\frac{t}{\tau_0}} \right] \tag{1}$$

$$\left[ \left( A_1 \cdot \left( e^{-\frac{t_m}{\tau_1}} - e^{-\frac{t_{max}}{\tau_1}} \right) \right) + \left( (A_2) \cdot \left( e^{-\frac{t_m}{\tau_1}} - e^{-\frac{t_{max}}{\tau_1}} \right) \right) \right]^{-1} \cdot \left[ \frac{A_1}{\tau_1} \cdot e^{-\frac{t}{\tau_1}} + \frac{A_1}{\tau_2} \cdot e^{-\frac{t}{\tau_2}} \right] \tag{2}$$

$$\text{where } A_1 + A_2 = 1$$

To judge the goodness of the fits, the log likelihood ratio test was used to determine if the simplest model (single-exponential distribution) was sufficient to describe the data (*Kaur et al., 2019*).

Kinetic modeling of RNA-4+2 was performed using QuB (*Nicolai and Sachs, 2013*) as previously described (*White et al., 2021*). Three different hidden Markov models were built, and the transition rates were globally optimized across all molecules using maximum idealized point estimation (*Qin et al., 2000*; *Figure 1—source data 3*). The goodness of fit of each model was assessed by the Bayesian information criterion (BIC) (*Schwarz, 1978*) in *Equation 3* where k is the number of free parameters in the model, N is the number of data points (i.e., frames) and LL is the log likelihood of the fit returned by QuB. The model with the lowest average BIC score across a fivefold resampling of the data was considered the best fit.

$$BIC = k \times \ln(N) - 2 \times \text{LL} \tag{3}$$

## Acquisition and analysis of higher frame rate data

To ensure that the lifetimes measured in these experiments are not limited by our acquisition rate of of 0.2 Hz, additional U1 snRNP experiments were performed with a continuous exposure of the Cy3 signal at 1 Hz for select RNAs (RNA-10, RNA-4+2, RNA-C, *Figure 1—figure supplement 4*). These experiments were performed on a second custom-built, objective-based micromirror total internal fluorescence microscope. Data acquisition and analysis were carried as described above with the following modifications. Laser powers were set between 800 and 2000 µW for 633 nm and 1000 µW for 532 nm. Excitation and emission passed through a 60×1.49 NA oil immersion

objective (Olympus). Emission was split with using a dual-view system built as previously described except that the optics were mounted in an optical cage (*Larson et al., 2014*). The images were projected onto two separate 2048×2048 sCMOS detectors (Hamamatsu ORCA-Flash4.0 V3) with 2×2 pixel binning. Imaging was controlled with Micro-Manager 2.0 (*Edelstein et al., 2014*). For these experiments, cleaned cover glasses were passivated with mPEG-SVA (MPEG-SVA-5K, Laysan Bio) and mPEG-biotin-SVA (BIO-PEG-SVA-5K, Laysan Bio) at a ratio of 1:100 w/w in 100 mM NaHCO$_3$ (pH 8) overnight. Following passivation, slides were rinsed with PBS, incubated with PBS +1 mg/mL bovine serum albumin (BSA) for 30 min, and rinsed with Mock Splicing Buffer supplemented with 1 mg/mL BSA. Videos were sequentially collected at 633 nm for 30 frames to identify surface-tethered U1 snRNP molecules followed by 532 nm excitation for 30 min (1800 frames) at 1 Hz to monitor the Cy3 channel. All 532 nm videos were background subtracted in ImageJ (version 2.1.0). Data analysis was performed using custom code written in MATLAB. The 633 and 532 nm channels were aligned using a similarity transform computed from images containing fluorescent beads (Life Technologies). U1 snRNP molecules were detected using a generalized log likelihood ratio test (*Sergé et al., 2008*) and locations were refined with a two-dimensional Gaussian. Drift correction was performed by computing and applying a similarity transform every 10 frames which tracked the location of fluorescent beads on the surface. The time-dependent fluorescence intensity in each channel was integrated over a 3×3 pixel space for each frame. Events in each time series were detected using the DISC algorithm (*White et al., 2020*) and visually inspected to ensure only specific binding events were included in the analysis.

## Acknowledgements

We thank David Brow, Sam Butcher, Joshua Larson, Margaret Rodgers, and Tucker Carrocci for critical reading of the manuscript. We thank Clarisse van der Feltz and Daniel Pomeranz Krummel for assistance in U1 snRNP purification. Funding: This work was supported by the National Institutes of Health (R01 GM 122735 and R35 GM136261 to AAH, R35 GM126914 to LMS and MS, and F32 GM143780 to DSW). SRH was supported in part by the NIH Chemistry-Biology Interface Training Grant (T32 GM008505).

## Additional information

### Competing interests

Ivan R Corrêa: is employed by New England Biolabs. Aaron A Hoskins: is conducting sponsored research with and a scientific advisor for Remix Therapeutics, Inc. The other authors declare that no competing interests exist.

### Funding

| Funder | Grant reference number | Author |
|---|---|---|
| National Institutes of Health | R01 GM122735 | Aaron A Hoskins |
| National Institutes of Health | R35 GM136261 | Aaron A Hoskins |
| National Institutes of Health | R35 GM126914 | Lloyd M Smith |
| National Institutes of Health | T32 GM008505 | Sarah R Hansen |
| National Institutes of Health | F32 GM143780 | David S White |

The funders had no role in study design, data collection and interpretation, or the decision to submit the work for publication.

## Author contributions
Sarah R Hansen, Conceptualization, Resources, Data curation, Software, Formal analysis, Validation, Investigation, Visualization, Methodology, Writing – original draft, Writing – review and editing; David S White, Software, Formal analysis, Validation, Investigation, Methodology, Writing – review and editing; Mark Scalf, Resources, Formal analysis, Methodology, Writing – review and editing; Ivan R Corrêa, Lloyd M Smith, Resources, Writing – review and editing; Aaron A Hoskins, Conceptualization, Resources, Supervision, Funding acquisition, Writing – original draft, Project administration, Writing – review and editing

## Author ORCIDs
David S White ⓘD http://orcid.org/0000-0003-0164-0125
Ivan R Corrêa ⓘD http://orcid.org/0000-0002-3169-6878
Lloyd M Smith ⓘD http://orcid.org/0000-0002-6652-8639
Aaron A Hoskins ⓘD http://orcid.org/0000-0002-9777-519X

## Decision letter and Author response
Decision letter https://doi.org/10.7554/eLife.70534.sa1
Author response https://doi.org/10.7554/eLife.70534.sa2

# Additional files

## Supplementary files
• Transparent reporting form

## Data availability
Source data files have been provided for Figure 1-Supplemental Figure 2. The source data for the single molecule microscopy experiments are hosted at Figshare, via the link https://doi.org/10.6084/m9.figshare.c.6164067. We have included this link in the manuscript text with the materials and methods section describing single molecule data collection.

The following dataset was generated:

| Author(s) | Year | Dataset title | Dataset URL | Database and Identifier |
|---|---|---|---|---|
| Hoskins A, Hansen SR, White DS, Scalf M, Correa IR, Smith LM | 2022 | Multi-step recognition of potential 5' splice sites by the Saccharomyces cerevisiae U1 snRNP | https://doi.org/10.6084/m9.figshare.c.6164067.v1 | Figshare, 10.6084/m9.figshare.c.6164067.v1 |

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
