## [Editor Report]

This study extends previous work from the same group on the mechanism of 5' splice site recognition by the U1 snRNP using co-localization single-molecule spectroscopy. Compelling experimental and analytical approaches yielded three important conclusions: (1) the association of the U1 snRNP with the 5' splice site is largely determined by the snRNP itself and does not require other splicing factors; (2) sequence features of the 5' splice site determine whether a short-lived complex with U1 dissociates or transitions into a longer-lived, "productive" complex, potentially mediated by stabilized contacts with U1 associated proteins; and (3) the ability to form the longer-lived complex cannot be accurately predicted by base-pairing potential alone, as presumed by many predictive algorithms. This work will be of interest to colleagues in the splicing field as well as to others in fields where nucleic acid recognition by snRNPs plays a major role.

---

## [Decision Letter]

**Decision letter after peer review:**

Thank you for submitting your article "Multi-Factor Authentication of Potential 5' Splice Sites by the *Saccharomyces cerevisiae* U1 snRNP" for consideration by *eLife*. Your article has been reviewed by 3 peer reviewers, including Jonathan P Staley as the Reviewing Editor and Reviewer #1, and the evaluation has been overseen by Kevin Struhl as the Senior Editor. The following individual involved in review of your submission has agreed to reveal their identity: Nils Walter (Reviewer #2).

Essential revisions:

1) The authors implicate protein in favoring the long-lived complex studied, imposing asymmetry of importance in base pairing, decreasing the stability of base pairing, and favoring base pairing length over thermodynamic stability. The authors further speculate a role for protein-RNA contacts involving Ych1 and Luc7, given informative, published structures. To corroborate the importance of protein in these features of 5' splice site recognition and to sharpen the mechanistic focus of the manuscript, the authors need to test the impact of Yhc1 and Luc7 mutants at the protein-RNA interface for roles in these features – especially Yhc1, given that the authors have already published on the impact of mutations in Yhc1. Otherwise, given the extensive work the authors have previously performed to already demonstrate that U1 snRNP binds to a 5'SS reversibly, with fast and slow dissociation events, one could argue that the current work falls somewhat short in providing new major biological insights.

2) On a related point, in the section describing U1/5'SS duplexes destabilization in U1 snRNP (line 281) an underlying assumption is that the binding of two RNAs (in the absence of the spliceosomal proteins) would share the same characteristics or trends as two identical RNAs incorporated into the U1 snRNP. While this may be a rhetorical device to increase the clarity/connection between the concepts of predicted binding free energies and the residence time of hybridized oligonucleotides, it does not address the possible reasons for the discrepancy observed in RNA oligonucleotide versus U1 snRNP binding. Further, in the Discussion (lines 395-398), the authors mention that while this study cannot identify a specific interaction or event that stabilizes the long-lived complex, structural studies implicate two U1 associated proteins: Yhc and Luc7. They further describe the interactions that could be implicated based on their findings. It is very difficult to follow this description of the contacts in the context of the larger snRNP structure without an illustrative figure. The authors should point to a reference and derive a physical model from the available cryo-EM structures to show that the U1 snRNA is, most likely, being constrained by its associated proteins in such a way that it increases the binding affinity to complementary RNA oligonucleotides. It would be helpful to add a figure based on the plethora of existing structural data to contextualize the findings of the current work (U1 SSRS/5'SS duplex), showing the protein contacts that the authors implicate in the conformational and thermodynamic modulation of the U1 SSRS/5'SS duplex.

3) Since splice sites are often "found" in the context of alternative or pseudo/near-cognate splice sites, it would be relevant to the biological significance of the study to ask whether the "rules" identified in the experiments presented in this study influence splice site competition and whether both the short- and long-lived states are subject to competition or, rather, only the short-lived complexes. If possible, it would be beneficial to repeat the CoSMoS experiment with two oligomer sequences of different colors or to assess the impact of adding an unlabeled competitor.

4) While the two-factor authentication metaphor of Figure 7 is charming, it seems off-topic. Instead, the authors should review the literature for examples of short, exploratory binding events involving an RNA:protein complex, followed by more stable, accommodated binding events, see e.g., the work by Sarah Woodson on 30S ribosomal subunit assemble and on Hfq function, work on kinetic proofreading of the ribosome, work on Cas9-based recognition of its target site, and many others. A potential descriptive framework to be used here is that of "conformational proofreading". Further, the use of "multi-factor authentication" seems inappropriate for a research article title.

5) The model described in the paragraphs starting with line 262 through 280 to interpret the observation of long and short complex lifetimes is not entirely clear. There are at least two potential models that can be considered to fit the observations: a linear and a circular model. A linear model would be one where U1 and substrate RNA are not associated (state 1), then they partially associate (state 2), and finally they isomerize to the completely associated/fully hybridized complex (state 3). The circular model is the same, except that it would additionally allow switching between states 1 and 3 directly (bypassing the partially associated state). To differentiate between these two scenarios, the authors would have to vary the concentration of the RNA probe and see if there is a uniform change in a single k_on_ rate or if two k_on_ rates start to appear. These rate subpopulations would be much easier to detect by fitting with hidden Markov models. It would seem unjustified to decide between these two models without obtaining such additional supporting data.

6) There is significant concern that the single molecule sampling rate used to acquire the CoSMoS data is too slow to accurately measure the shortest lifetimes observed, which are only ~10 seconds long. According to the Nyquist sampling criterion, the sampling rate needs to be (at least) twice the frequency of the event being measured, implying that the authors cannot meaningfully observe any lifetime shorter than ~10 seconds given their limited sampling rate. Further considering that at minimum two consecutive data points are needed for observing a 10 second lifetime, artifacts (e.g., camera noise) could make up a disproportionate amount of the signal observed in their data for these short lifetimes. For an accurate measurement, the authors need to repeat the experiments at a higher sampling rate to make sure that there are no faster, transient interactions than those currently reported, and that the values reported are accurate.

7) The authors have chosen to extrapolate rates via exponential fitting to dwell time distributions. This is a reductive approach that ignores the relationship between consecutive events. It is strongly recommended that the authors consider using a hidden Markov modeling (HMM) approach instead. HMMs have long become the gold standard in single molecule biophysics. Even better, a Bayesian approach could help analyze entire datasets at the same time. In this reviewer's opinion, the ebFRET software package from the Gonzalez lab at Columbia University could, for example, work well here.

8) The authors should say more about the particular requirement for basepairing at position 6, especially in the context of the experiments in Figure 5. This is particularly striking as this position is not well conserved in natural 5'ss, at least compared to position 5.

---

## [Author Response]

Essential revisions:1) The authors implicate protein in favoring the long-lived complex studied, imposing asymmetry of importance in base pairing, decreasing the stability of base pairing, and favoring base pairing length over thermodynamic stability. The authors further speculate a role for protein-RNA contacts involving Ych1 and Luc7, given informative, published structures. To corroborate the importance of protein in these features of 5' splice site recognition and to sharpen the mechanistic focus of the manuscript, the authors need to test the impact of Yhc1 and Luc7 mutants at the protein-RNA interface for roles in these features – especially Yhc1, given that the authors have already published on the impact of mutations in Yhc1. Otherwise, given the extensive work the authors have previously performed to already demonstrate that U1 snRNP binds to a 5'SS reversibly, with fast and slow dissociation events, one could argue that the current work falls somewhat short in providing new major biological insights.

We disagree with the reviewers that this work as is falls short of providing new major biological insights and will respond to this item first. First, a major question from our previous studies has been ambiguity surrounding the source of both long- and short-lived binding events observed between U1 and substrate RNAs. Previously we could not rule out the presence of either RNA secondary structure or unknown components of the extract as sources of these observations. Our work here with purified U1 shows that U1 itself can interact with substrate RNAs in kinetically distinct ways. This information is critical for understanding the fundamental basis of snRNP/RNA recognition.

Second, we show that binding behaviors are highly variable depending on the extent of base pairing and the position of mismatches. Importantly, strong positional effects lead to discrimination against mismatches at certain sites and not others even when the predicted thermodynamic stability for duplex formation is strong. While this has been implied by numerous other studies, our work shows that this discrimination has a kinetic basis. Finally, this discrimination appears to be occurring at a step after initial binding: RNAs with mismatches at certain cites still associate with U1 but do so transiently. This indicates a kinetic proofreading-like mechanism is at play that permits rapid and reversible surveillance of RNAs, efficient rejection of those containing certain mismatches, and efficient stabilization of those containing high complementary to U1. All these pieces of information are fundamental for understanding how splice site recognition occurs and for understanding the kinetic and thermodynamic basis of these events rather than just inferring rules based on phenomenological observations.

In response to the second point, developing purification protocols for U1 snRNPs containing mutant U1s is well beyond the scope of the current manuscript. First, relatively little is understood about how Luc7 impacts 5’ splice site usage beyond early studies by Mattaj and Seraphin and more recent work by Shuman and Schwer. Specifically, neither group has addressed how Luc7 mutations impact usage of sequences which is necessary to correlate protein mutations with impact on binding of some sequences and note others. Second, mutation of either Yhc1 or Luc7 may significantly destabilize the snRNP and prevent its purification. There is already precedent for this. The relatively benign (at 30^o^C) *luc7-1* mutation that impacts 5’ splice site usage dramatically alters the protein composition of the U1 snRNP purified using the TAP tag (see Fortes, Mattaj, et al., Genes & Dev, 1999; figure 5D). While this and other mutants are useful mechanistic tools, the experimental requirements for confirming a compositionally and structurally intact mutant U1 snRNPs are beyond the scope of the manuscript.

2) On a related point, in the section describing U1/5'SS duplexes destabilization in U1 snRNP (line 281) an underlying assumption is that the binding of two RNAs (in the absence of the spliceosomal proteins) would share the same characteristics or trends as two identical RNAs incorporated into the U1 snRNP. While this may be a rhetorical device to increase the clarity/connection between the concepts of predicted binding free energies and the residence time of hybridized oligonucleotides, it does not address the possible reasons for the discrepancy observed in RNA oligonucleotide versus U1 snRNP binding. Further, in the Discussion (lines 395-398), the authors mention that while this study cannot identify a specific interaction or event that stabilizes the long-lived complex, structural studies implicate two U1 associated proteins: Yhc and Luc7. They further describe the interactions that could be implicated based on their findings. It is very difficult to follow this description of the contacts in the context of the larger snRNP structure without an illustrative figure. The authors should point to a reference and derive a physical model from the available cryo-EM structures to show that the U1 snRNA is, most likely, being constrained by its associated proteins in such a way that it increases the binding affinity to complementary RNA oligonucleotides. It would be helpful to add a figure based on the plethora of existing structural data to contextualize the findings of the current work (U1 SSRS/5'SS duplex), showing the protein contacts that the authors implicate in the conformational and thermodynamic modulation of the U1 SSRS/5'SS duplex.

We agree with the Reviewer and have edited Figure 1 to include an illustrative figure of the yeast U1/5’SS interaction. These are now Figure 1a, b.

3) Since splice sites are often "found" in the context of alternative or pseudo/near-cognate splice sites, it would be relevant to the biological significance of the study to ask whether the "rules" identified in the experiments presented in this study influence splice site competition and whether both the short- and long-lived states are subject to competition or, rather, only the short-lived complexes. If possible, it would be beneficial to repeat the CoSMoS experiment with two oligomer sequences of different colors or to assess the impact of adding an unlabeled competitor.

The reviewer proposes an interesting experiment; however, at this stage do not believe it is possible to interpret such an experiment in terms of biological significance since competing 5’SS maybe present in different RNA structures or bound to different factors in vivo.

4) While the two-factor authentication metaphor of Figure 7 is charming, it seems off-topic. Instead, the authors should review the literature for examples of short, exploratory binding events involving an RNA:protein complex, followed by more stable, accommodated binding events, see e.g., the work by Sarah Woodson on 30S ribosomal subunit assemble and on Hfq function, work on kinetic proofreading of the ribosome, work on Cas9-based recognition of its target site, and many others. A potential descriptive framework to be used here is that of "conformational proofreading". Further, the use of "multi-factor authentication" seems inappropriate for a research article title.

We agree with the reviewer that these other examples are noteworthy, and, in fact, we already included references to many of these in the manuscript in the Discussion section titled General Features of Nucleic Acid Recognition by RNPs.

While we think the multifactor authentication is a useful framework for thinking about the events involved in splice site recognition, we value the reviewers concerns that this perhaps is too ambiguous in a biochemical context. We have altered the title of the manuscript to reflect this, changed the graphics for the model proposed in Figure 7A, and have included a discussion of conformational proofreading on pages 19-20. We did not reference ribosome assembly work by Woodson but have now included that in revised manuscript in our discussion of conformational proofreading, which is provide below for convenience.

“Combined, our data also indicate that binding involves several checkpoints before long-lived complexes are formed (Figure 7). This scheme is consistent with 5' SS recognition involving conformational proofreading as has been implicated in many other nucleic acid recognition events including ribosome assembly and translation (Rodgers and Woodson, 2019, Blanchard et al., 2004). In the case of U1, the conformational change that leads to proofreading (rapid release of the RNA) or stable binding may involve rearrangement of Yhc1 and Luc7 as discussed in the preceding paragraph. Thus, the conformational proofreading would involve formation of different states of the U1 snRNP with different kinetic properties. The initial barrier to forming short-lived complexes is low and requires limited complementarity to the SSRS. Formation of this complex is readily reversible which permits rapid surveillance of transcripts by U1 for 5' SS and prevents accumulation of U1 on RNAs lacking features necessary for splicing.”

5) The model described in the paragraphs starting with line 262 through 280 to interpret the observation of long and short complex lifetimes is not entirely clear. There are at least two potential models that can be considered to fit the observations: a linear and a circular model. A linear model would be one where U1 and substrate RNA are not associated (state 1), then they partially associate (state 2), and finally they isomerize to the completely associated/fully hybridized complex (state 3). The circular model is the same, except that it would additionally allow switching between states 1 and 3 directly (bypassing the partially associated state). To differentiate between these two scenarios, the authors would have to vary the concentration of the RNA probe and see if there is a uniform change in a single k_on_ rate or if two k_on_ rates start to appear. These rate subpopulations would be much easier to detect by fitting with hidden Markov models. It would seem unjustified to decide between these two models without obtaining such additional supporting data.

This is good point. Collecting more data with RNA-4+2 would be a great way to get address this question; however, our data were already collected at 10 nM RNA which is the effective concentration barrier in these experiments due to background fluorescence and non-specific binding of the oligos that resulted in data too noisy to reliably interpret.

We have performed a new analysis to compare the probability of different models for the current data. We have built and optimized three different kinetics models in QuB using maximum idealized point estimation and ranked them according to their Bayesian information criterion (weighting the likelihood of the model with the number of free parameters in the model) for RNA-4+2 binding to U1 snRNP. This is an HMM approach that directly optimizes rate constants from idealized data for a user constructed model (see Qin, Auerbach, and Sachs, Biophysical Journal, 2000).

These new results are now provided in Figure 1G (see comment #2), and Figure 1-Source Data 3.

Overall, Model 2 featuring a sequential RNA association followed by a longer-lived bound state provided the lowest BIC score which suggests this model best explains our data (of the models we have tested). Model 3 on the other hand, a circular version of Model 2 proposed by the reviewer, is less favorable for explaining our observations. These results suggest the additional state transitions in Model 3 are not necessary to describe our observations. We are confident in the model we have proposed in the manuscript, given the limitations of our data.

Of note, we have performed this analysis on RNA-4+2 as this sequence accounts for a very common 5’SS in yeast (~26% of yeast introns (77/298) are G/GUAUGU) as well as the 5’SS observed in the most frequently studied model introns (RP51A, ACT1, and UBC4). We do not wish to over interpret our data with respect to other, less common splice sites.

A new section to the Methods has been added to describe our HMM methods, which is provided below for convivence.

“Kinetic modeling of RNA-4+2 was performed using QuB (Nicolai and Sachs, 2013) as previously described (White et al., 2021). Three different hidden Markov models were built, and the transition rates were globally optimized across all molecules using maximum idealized point estimation (Qin et al., 2000). (Figure 1-Source Data 3). The goodness of fit of each model was assessed by the Bayesian information criterion (Schwarz et al., 1978) in Equation 3 where *k* is the number of free parameters in the model, *N* is the number of data points (i.e., frames) and *LL* is the loglikelihood of the fit returned by QuB. The model with the lowest average BIC score across a five-fold resampling of the data was considered the best fit.

BIC=kxln(N)−2xLL
*(3)*

6) There is significant concern that the single molecule sampling rate used to acquire the CoSMoS data is too slow to accurately measure the shortest lifetimes observed, which are only ~10 seconds long. According to the Nyquist sampling criterion, the sampling rate needs to be (at least) twice the frequency of the event being measured, implying that the authors cannot meaningfully observe any lifetime shorter than ~10 seconds given their limited sampling rate. Further considering that at minimum two consecutive data points are needed for observing a 10 second lifetime, artifacts (e.g., camera noise) could make up a disproportionate amount of the signal observed in their data for these short lifetimes. For an accurate measurement, the authors need to repeat the experiments at a higher sampling rate to make sure that there are no faster, transient interactions than those currently reported, and that the values reported are accurate.

This is a valid concern. We have performed new single molecule experiments at faster collection rate (1 frame per second, 5x increase in temporal resolution) for RNA-10, RNA-4+2, and RNA-C. We did not find any evidence for faster, transient interactions. These new data are now included as Figure 1-Supplement 4 and Figure 1- Source Data 2 with the corresponding changes to the Main text and Methods.

Addition to Methods in the section titled “Acquisition and Analysis of Higher Frame Rate Data”

“To ensure that the lifetimes measured in these experiments are not limited by our acquisition rate of 0.2 Hz, additional U1 snRNP experiments were performed with a continuous exposure of the Cy3 signal at 1 Hz for select RNAs (RNA-10, RNA-4+2, RNAC, Figure 1—figure supplement 4) […] Events in each time series were detected using the DISC algorithm (White et al., 2020) and visually inspected to ensure only specific binding events were included in the analysis.”

We do note however that in these data sets, we are able to observe the short-lived (~40s) binding events with RNA-10. We have revisited our data collected at 0.2 Hz (1 frame every 5 seconds) and indeed find evidence of a short- and long-lived state. As such, we have updated Figure 3, the corresponding supplemental tables, and the discussion of data with RNA-10 accordingly.

7) The authors have chosen to extrapolate rates via exponential fitting to dwell time distributions. This is a reductive approach that ignores the relationship between consecutive events. It is strongly recommended that the authors consider using a hidden Markov modeling (HMM) approach instead. HMMs have long become the gold standard in single molecule biophysics. Even better, a Bayesian approach could help analyze entire datasets at the same time. In this reviewer's opinion, the ebFRET software package from the Gonzalez lab at Columbia University could, for example, work well here.

We disagree with this statement. While HMMs are powerful, we do not believe they are the “gold standard” for all single molecule measurements in biophysics. Alternative approaches including change-point detection, auto/cross correlation analysis, image recognition, and others are commonly used in biophysics, particularity for “model-free” analysis. For certain cases, ebFRET is indeed a powerful tool; however, this software was designed specifically for intramolecular smFRET data, and it is unclear which priors are appropriate for intermolecular CoSMoS data without further investigations that are beyond the scope of this work.

Herein, events were discretized via thresholding and further inspected by examining the raw camera data at each AOI to discern specific and non-specific binding, a well described and robust approach for CoSMoS analysis (Friedman and Gelles, 2015) The exact classification of *unbound* and *bound* events is unlikely to change between our method and a standard HMM given the high signal to background of our data (see example traces in Figure 1 and 3). An example of this is clearly provided in Figure 3 of Greenfield et al., PLOS One, 2012 where thresholding and HMMs provided similar results at high signal to noise ratios in simulated smFRET data. In fact, our association rates reported in Figure 2-Source Data 1 are free of contamination from photobleaching (a feature often not corrected for in HMM modeling) as we only analyzed the time to the first association event. Therefore, our analysis in the main text is appropriate.

For our new, faster framerate data at 0.2 Hz, the signal to noise ratio is lower, as this set up uses sCMOS detectors. Therefore, these data were analyzed by the DISC algorithm (White et al., 2020), which has exhibited higher accuracy for intermolecular CoSMoS data than vbFRET (the predecessor of ebFRET). DISC is like the segmental k-means algorithm (SKM) and uses unsupervised learning to approximate an HMM. The final output of DISC (and SKM) is the Viterbi path of state transitions (i.e., maximum probability assignment of each data point into one of the discrete states given the determined transition and emission matrices). Importantly, our new analysis of RNA-10 and RNA-4+2 provided dwell time distributions with similar parameters from MLE fitting, suggesting both analysis methods are providing consistent results (see reviewer Comment #6).

To the first point of the reviewer, we agree more information can indeed be obtained by considering the time between successive events rather than dwell time analysis alone. We have now performed HMM analysis for RNA-4+2 which represents a very common 5’SS observed in vivo to compare three specific models (see response to Comment #5). For our survey of >20 other RNAs, we have chosen to only compare maximum likelihood estimates (MLE) of dwell time distributions as we are not confident a parsimonious model can be selected for each dataset. Thus, to avoid over interpreting our data, our manuscript focuses on a discussion of parameters from MLE fitting, a long standing a reliable approach in the single-molecule community.

8) The authors should say more about the particular requirement for basepairing at position 6, especially in the context of the experiments in Figure 5. This is particularly striking as this position is not well conserved in natural 5'ss, at least compared to position 5.

Position +6 (G/GUAUGU) is frequently conserved as a “U” in both yeast (as we show in Figure 7B) and in humans (see Roca, Sachidanandam, and Krainer, RNA, 2005). In fact, of 245 “GUAUG” introns in yeast, 227 (93%) of them also contain a U at +6 (GUAUGU). Presumably, this site is conserved since it permits pairing with the U6 snRNA “ACAGA” sequence during the catalytic steps in splicing. We have expanded on this idea in the discussion and this text is included on page 21.

“An additional consequence of this length requirement and the duplexes described in Figure 7C is that they also favor base pairing interactions between the 5' end of the U1 SSRS and the 3' end of the splice site. For example, each of the RNAs in our study capable of forming long-lived interactions also could pair at the +6 position (G/GUAUGU) of the 5' SS. This particular position of the 5' SS is important since it also pairs with the “ACAGA” sequence of the U6 snRNA (base-pairing position underlined) to promote splicing catalysis (Sontheimer and Steitz, 1993, Kandles-Lewis and Seraphin, 1993, Kim and Abelson, 1996) As mentioned above for recognition of G+1, the kinetic properties of U1 are, in part, optimized to facilitate interactions between the 5' SS and the splicing machinery that are important for catalysis even after U1 is released.”